# Clients Collaborate: Flexible Differentially Private Federated Learning with Guaranteed Improvement of Utility-Privacy Trade-off

Yuecheng Li[1]  Lele Fu[1]  Tong Wang[2]  Jian Lou[1]  Bin Chen[3]  Lei Yang[1]  Jian Shen[4]  Zibin Zheng[1]
Chuan Chen[1]

## Abstract

To defend against privacy leakage of user data, differential privacy is widely used in federated learning, but it is not free. The addition of noise randomly disrupts the semantic integrity of the model and this disturbance accumulates with increased communication rounds. In this paper, we introduce a novel federated learning framework with rigorous privacy guarantees, named **FedCEO**, designed to strike a trade-off between model utility and user privacy by letting clients "*Collaborate with Each Other*". Specifically, we perform efficient tensor low-rank proximal optimization on stacked local model parameters at the server, demonstrating its capability to *flexibly* truncate high-frequency components in spectral space. This capability implies that our FedCEO can effectively recover the disrupted semantic information by smoothing the global semantic space for different privacy settings and continuous training processes. Moreover, we improve the SOTA utility-privacy trade-off bound by order of $\sqrt{d}$, where $d$ is the input dimension. We illustrate our theoretical results with experiments on representative datasets and observe significant performance improvements and strict privacy guarantees under different privacy settings. The **code** is available at https://github.com/6lyc/FedCEO_Collaborate-with-Each-Other.

## 1. Introduction

Federated learning (FL) (McMahan et al., 2017), a privacy-preserving distributed machine learning paradigm, enables multiple parties to jointly learn a model under global scheduling while keeping the data from leaving the local client. Nevertheless, recent work has shown that an attacker (i.e., the server or a particular client) can steal raw training data (Zhu et al., 2019; Jeon et al., 2021) and even specific private information (Fowl et al., 2022) by inverting parameters (gradients) uploaded from other clients. To further enhance privacy safeguards, differential privacy (DP) (Dwork, 2006; 2010) has become the prevailing standard in privacy-preserving machine learning (Jain et al., 2018; Levy et al., 2021). This privacy computing technique has proven effective in federated learning, guarding against client (user) privacy breaches by introducing random noise to uploaded updates (McMahan et al., 2018; Jain et al., 2021; Malekmohammadi et al., 2024). Unfortunately, randomized mechanisms like DP may result in a sacrifice of model utility, especially as the number of communication rounds increases (Yuan et al., 2023). Overall, the key issue is how to achieve an improved utility-privacy trade-off in differentially private federated learning (DPFL), which constitutes a novel and challenging research direction (Bietti et al., 2022; Cheng et al., 2022; Shen et al., 2023; Tsoy et al., 2024).

Considering the randomness of the introduced noise for differential privacy, the specific semantic information that gets corrupted can differ across individual clients. Meanwhile, there exists a certain similarity between the data distributions of each client, leading to the correlation in their semantic spaces. Therefore, we propose to mitigate the impact of DP noise on the utility of the global model in federated learning by exploiting the semantic complementarity between the noisy parameters of different clients. To substantiate our motivation, we visualize the smoothness of the global semantic space at different stages of training (see the heat map in Figure 1). Specifically, we compute Laplacian regularization values for the last linear layer of the backbone along the client-side direction, which serves as a metric for assessing the smoothness of the global semantic space (Yin et al., 2015; Pang & Cheung, 2017). Furthermore,

---

[1]Sun Yat-sen University, Guangzhou, China [2]Texas A&M University, Texas, USA [3]Harbin Institute of Technology, Shenzhen, China [4]Zhejiang Sci-Tech University, China. Correspondence to: Chuan Chen <chenchuan@mail.sysu.edu.cn>.

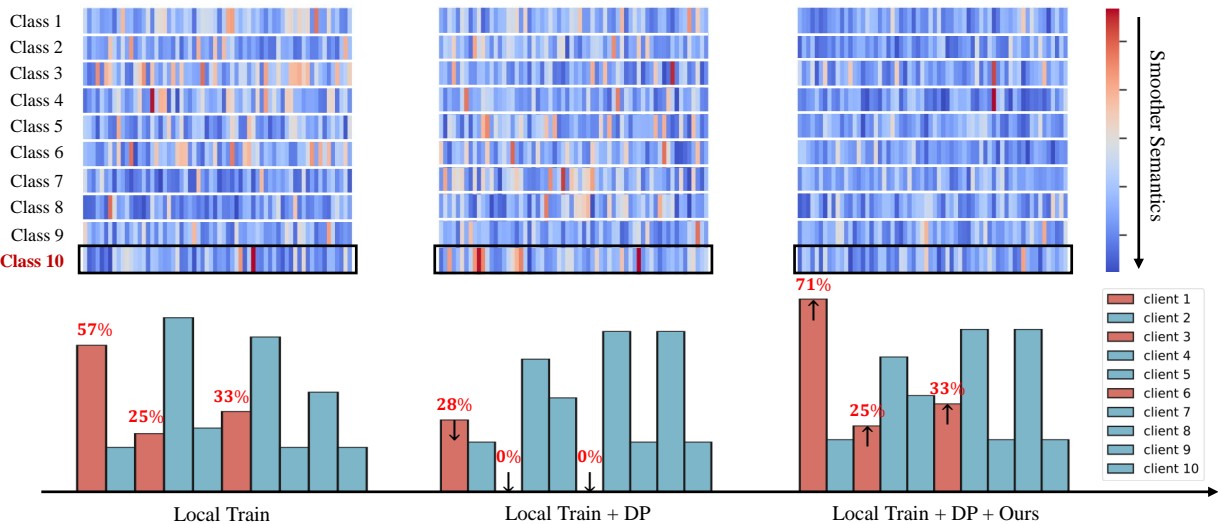

*Figure 1.* The heat map illustrates the smoothness of the global semantic space for ten clients at different training stages, where the color gradient from red to blue signifies an increase in smoothness. The bar chart illustrates the testing accuracy on the tenth class for each client during different training stages, where clients experiencing significant degradation in the semantic understanding of the tenth class are highlighted in pink, while others are marked in green. Due to the randomness of DP noise, we observe significant degradation in the local semantic representation of 10-th class for clients 1, 3, and 6 among the ten clients (ACC significantly reduced), while the impact on other clients is relatively minor. Consequently, the corresponding blocks in the tenth row of the heat map matrix turn red, indicating a decrease in the smoothness of the global semantic space. In contrast, our approach enhances the smoothness of the global semantic space, evidenced by the recovery of the blue color in the tenth row of the heat map matrix. Simultaneously, the local semantic representation of class ten for clients 1, 3, and 6 is restored (ACC improved), based on the collaboration among all clients.

we present the testing accuracy for each client concerning the 10-th class on the CIFAR-10 dataset with a two-layer multi-layer perceptron (MLP) (see the bar chart Figure 1). Taking the 10-th class in CIFAR-10 as an illustrative example, we observe that the introduction of differential privacy disrupts the smoothness of the global semantic space for the tenth class (as evidenced by the reddening of the tenth row). Simultaneously, there is a significant decline in testing accuracy for this class across clients 1, 3, and 6. Following our proposed low-rank processing, we observe an enhancement in the smoothness of the global semantic space for the tenth class (as evidenced by the bluing of the tenth row), concomitant with an increase in accuracy for clients 1, 3, and 6. Moreover, a similar phenomenon is noted for the seventh class (as observed in the seventh row of the heat map). This emphasizes that smoothing the global semantic space based on the semantic complementarity of noise parameters between clients is key to addressing the declining utility of models in the DPFL framework. While previous works have also focused on improving utility in DPFL, they are fundamentally based on restricting the local updates (Cheng et al., 2022; Shen et al., 2023) without considering the collaborative relationship across different clients. In other words, our work provides a **new perspective** on the utility-privacy trade-off in federated learning by exploring

an inter-client semantic collaboration approach.

**Contributions.** In this work, we propose an optimization algorithm in the server based on tensor low-rank techniques, which offers a *flexible* approach to smoothing the global semantic space for *different privacy settings* and *continuous training processes*. We provide rigorous theoretical analysis and extensive empirical evidence to substantiate our proposed framework. Specifically, our contributions are summarized as follows:

1. We introduce a novel federated learning framework, **FedCEO**, characterized by stringent privacy guarantees and enhanced model utility. We establish its equivalence in the spectral space to the truncated tensor singular value decomposition (T-tSVD) algorithm (Kilmer & Martin, 2011), showing its ability to achieve smoothness by truncating high-frequency components in the global semantic space. Benefiting from the T-tSVD, we can flexibly control the degree of semantic complementarity between clients according to noise levels.

2. We theoretically prove a new utility-privacy tradeoff bound for our FedCEO yielding a notable improvement of $O(\sqrt{d})$ over previous SOTA results due to low-rankness, where $d$ is the input dimension.

3. We empirically demonstrate that our model utility outperforms previous work under various model architectures and privacy settings. Furthermore, we also employ a common gradient inversion algorithm named *DLG* (Zhu et al., 2019) to attack our framework, validating its robust privacy-preserving performance. In summary, we demonstrate that our approach achieves the best trade-off between utility and privacy within the framework of DPFL.

## 1.1. Related Work

To ensure formal privacy guarantees, federated learning with differential privacy has undergone extensive investigation (Kim et al., 2021; Naseri et al., 2022). Recent efforts have been concentrated on user-level differential privacy, aiming to enhance model utility (McMahan et al., 2018; Wei et al., 2021; Shen et al., 2023). In alignment with these endeavors, we also adopt user-level differential privacy and employ it locally to safeguard against potential adversarial threats to the server (Fowl et al., 2022; Hu et al., 2024). Moreover, to further enhance the model utility of DPFL, current methods mainly investigate techniques such as regularization or personalization, fundamentally constraining the size of uploaded updates. (Cheng et al., 2022) proposes two regularization techniques: bounded local update regularization and local update sparsification. These methods enforce constraints to reduce the norm of local updates. In a more natural paradigm, PPSGD (Bietti et al., 2022) introduces a personalized privacy-preserving stochastic gradient optimization algorithm designed for training additive models with user-level differential privacy. It decomposes the model into the sum of local and global learning components, selectively sharing only the global part. However, extending this method to more complex personalized models proves challenging. Work closely related to ours are (Jain et al., 2021) and CENTAUR (Shen et al., 2023), who perform singular value decomposition (SVD) on noisy representation matrices of clients individually at the server. They verify that handling such issues in the spectral space is promising. However, their methods independently explore spectral information. In contrast, we stack the noisy models into a higher-order tensor, capitalizing on the semantic complementarity among clients to further enhance model utility in DPFL. Consequently, we improve the utility-privacy trade-off from their $O(d^{1.5})$ and $O(d)$ to our $O(\sqrt{d})$.

## 2. Preliminaries

### 2.1. Federated Learning

Federated learning (FL) (McMahan et al., 2017) is a multi-round protocol involving a central server and a set of local clients collaboratively training a privacy-preserving global model. In the federated learning, we address the following formulation of the optimization problem:

$$\min_{w \in R^d} \left\{ f(w) := \frac{1}{K} \sum_{k \in S_t} f_k(w) \right\}, \qquad (1)$$

where $N$ is the total number of clients, with a selection of a subset $S_t$ of size $K$ in each round. Let $w$ denote the parameters of the global model, and $D_k$ denote the local dataset of client $k$. Let $f_k := \mathbb{E}_{b \sim D_k}[F(w; b)]$ denote the empirical risk for client $k$, where $b$ represents a randomly sampled mini-batch from the local dataset $D_k$.

To achieve the goal of collaborative training without exposing local data, federated learning employs parameter transmission followed by aggregation. However, FL introduces challenges not encountered in centralized learning, such as gradient conflicts arising from data heterogeneity and privacy risks due to gradient leakage. Our paper primarily focuses on privacy concerns in FL and explores further utility guarantees.

### 2.2. Differential Privacy

Differential Privacy (DP) is a privacy computing technique with formal mathematical guarantees. We commence by presenting the definition of base DP.

**Definition 2.1** (**Differential Privacy (Dwork, 2006; Abadi et al., 2016)**)**.** A randomized mechanism $\mathcal{M} : \mathcal{D} \to \mathcal{R}$ with data domain $\mathcal{D}$ and output range $\mathcal{R}$ gives $(\epsilon, \delta)$-differential privacy if for any two adjacent datasets $D, D' \in \mathcal{D}$ that differ by *one record* (add or remove) and all outputs $S \subseteq \mathcal{R}$ it holds that

$$\Pr[\mathcal{M}(D) \in S] \le e^\epsilon \Pr[\mathcal{M}(D') \in S] + \delta,$$

where $\epsilon$ and $\delta$ denote the privacy budget and the relaxation level, respectively. It indicates the hardness of obtaining information about one record in the dataset by observing the output of the algorithm $\mathcal{M}$, especially when the privacy budget $\epsilon$ is smaller.

**User-level DP**: In this paper, we employ **user-level differential privacy**, which naturally suits federated learning, shifting the protected scale to individual user (Dwork, 2010; McMahan et al., 2018). In other words, *one record* in Definition 2.1 consists of all data belonging to a single client.

### 2.3. Differential Privacy in Federated Learning

User-level differential privacy provides formal privacy guarantees for individual clients and has found widespread applications in federated learning. (McMahan et al., 2018) first introduces DP-FedAvg and DP-FedSGD, formally incorporating user-level DP into the FL framework. These methods involve *clipping* per-user updates at the client side, followed by aggregation and the addition of *Gaussian noise* at the

**Algorithm 1** UDP-FedAvg

**Input:** $K$: number of participating clients each round, $T$: communication rounds, $C$: clipping threshold, $\sigma$: noise multiplier, $\eta$: learning rate, $E$: local epochs.
**Output:** $w'(T)$: global model.

1: Initialize global model $w(0)$ randomly
2: **for** $t = 1$ **to** $T$ **do**
3:     Take a random subset $S_t$ of $K$ clients (Total $N$)
4:     **for** all clients $k$ **in parallel do**
5:         $w'_k(t) = $ **ClientDPUpdate** $(w'(t-1), k)$
6:     **end for**
7:     $w'(t) = \frac{1}{K} \sum_{k \in S_t} w'_k(t)$
8:     return $w'(t)$
9: **end for**
10: return $w'(T)$

**ClientDPUpdate**$(w_0, k)$

1: Initialize $w = w_0$ (i.e. $w'(t-1)$)
2: **for** $e = 1$ **to** $E$ **do**
3:     Take a random split $\mathcal{B}$ from local dataset $D_k$ (i.e. mini-batches with size $B$)
4:     **for** batch $b \in \mathcal{B}$ **do**
5:         $w = w - \eta \frac{1}{B} \left( \sum_{(\boldsymbol{x},\boldsymbol{y}) \in b} \nabla_w F(w; \boldsymbol{x}, \boldsymbol{y}) \right)$
6:     **end for**
7: **end for**
8: $\Delta = w - w_0$
9: $\tilde{\Delta} = $ **GradientClip**$(\Delta)$
10: $w' = w_0 + \eta \left( \tilde{\Delta} + \mathcal{N}(0, \boldsymbol{I}\sigma^2 C^2 / K) \right)$
11: return $w'$

**GradientClip**$(\Delta)$

1: $\tilde{\Delta} = \Delta / \max\left(1, \frac{\|\Delta\|}{C}\right)$
2: return $\tilde{\Delta}$

server side, which ensures privacy protection for "large step" updates derived from user-level data. To further address potential malicious actions at the server (Fowl et al., 2022), we introduce user-level differential privacy (UDP) locally (Truex et al., 2020), as shown in Algorithm 1. Although the aforementioned algorithms provide user-level privacy guarantees in federated learning, they often result in a significant degradation of model utility. In our work, we specifically focus on strategies to enhance the utility of FL models while maintaining stringent privacy guarantees.

## 2.4. Low-rankness

The low-rank property is a crucial characteristic of both natural and artificial data, offering a description of dependence relationships among different dimensions. For instance, in the case of a second-order matrix, low-rankness character-

izes the correlation between elements in rows or columns, finding extensive applications in areas such as image denoising (Ren et al., 2022) and data compression (Idelbayev & Carreira-Perpinán, 2020). This paper focuses primarily on the low-rank property of third-order tensors, employing tensor nuclear norm (TNN) (Yang et al., 2016; Lu et al., 2016) for characterization and the tensor singular value decomposition (tSVD) (Kilmer & Martin, 2011; Lu et al., 2019) algorithm for modeling. The following sections provide a detailed introduction to these relevant concepts.

The Discrete Fourier Transform (DFT) is implicated in several related concepts introduced later in this paper. We denote discrete Fourier transform as $\mathrm{DFT}(\cdot)$, and more details are deferred to the **Appendix** A.1.

For each third-order tensor $\mathcal{W}$, we perform the discrete Fourier transform along its third dimension, denoted as $\overline{\mathcal{W}} = \mathrm{DFT}(\mathcal{W}, 3)$, where $\overline{\boldsymbol{W}}^{(i)}$ represents the $i$-th frontal slice of $\overline{\mathcal{W}}$. Similarly, we have $\mathcal{W} = \mathrm{IDFT}(\overline{\mathcal{W}}, 3)$, indicating the inverse discrete Fourier transform along the third dimension.

### 2.4.1. TENSOR NUCLEAR NORM

The Tensor Nuclear Norm (TNN) is often employed in optimization problems as a metric for the low-rank property of a tensor. A smaller nuclear norm indicates a lower rank for the tensor, implying stronger smoothness among its slices.

**Definition 2.2** (**Tensor Nuclear Norm (Lu et al., 2016)**). For each third-order tensor $\mathcal{W} \in \mathbb{R}^{n_1 \times n_2 \times n_3}$, it defines the tensor nuclear norm, denoted as $\|\cdot\|_*$, as follows:

$$\|\mathcal{W}\|_* := \frac{1}{n_3} \sum_{i=1}^{n_3} \left\|\overline{\boldsymbol{W}}^{(i)}\right\|_*.$$

It means the average of the matrix nuclear norm of all the frontal slices of $\overline{\mathcal{W}}$, where $\overline{\mathcal{W}}$ is a tensor after DFT on $\mathcal{W}$ along the third dimension.

### 2.4.2. T-PRODUCT AND TENSOR SVD

The tensor singular value decomposition (tSVD) can be employed to approximate low-rank tensors. To do so, we first introduce the concept of the tensor-tensor product (t-product), denoted as $\mathcal{U} * \mathcal{V}$ (Kilmer & Martin, 2011). More details are deferred to **Appendix** A.2.

Based on the t-product, we define tensor singular value decomposition (tSVD) as follows.

**Definition 2.3** (**Tensor Singular Value Decomposition (Kilmer & Martin, 2011; Lu et al., 2019)**). For each third-order tensor $\mathcal{W} \in \mathbb{R}^{n_1 \times n_2 \times n_3}$, it can be factored in

$$\mathcal{W} = \mathcal{U} * \mathcal{S} * \mathcal{V}^H,$$

**Algorithm 2** FedCEO

    **Input:** $K$: number of participating clients each round,
    $T$: communication rounds, $C$: clipping threshold,
    $\sigma$: noise multiplier, $\eta$: learning rate, $E$: local epochs,
    $I$: interval, $\lambda$: initial coefficient, $\vartheta$: common ratio.
    **Output:** $w'(T)$: global model.
1:  Initialize global model $w(0)$ randomly
2:  **for** $t = 1$ **to** $T$ **do**
3:     Take a random subset $S_t$ of $K$ clients (each client $k$
      is selected with probability $p = \frac{K}{N}$)
4:     **for** all clients $k$ **in parallel do**
5:       **if** $(t-1)\%I == 0$ **then**
6:         $w'_k(t) = $ **ClientDPUpdate** $\left(\hat{w}_k\left(t-1\right), k\right)$
7:       **else**
8:         $w'_k(t) = $ **ClientDPUpdate** $\left(w'\left(t-1\right), k\right)$
9:       **end if**
10:    **end for**
11:   **if** $t\%I == 0$ **then**
12:     $\mathcal{W}_{\mathcal{N}} = \text{fold}\left(\left[w'_1(t), \cdots, w'_K(t)\right]^T\right)$
13:     $\hat{\mathcal{W}} = \arg\min_{\mathcal{W}} \left\{\lambda/\vartheta^{\frac{t}{I}} \|\mathcal{W} - \mathcal{W}_{\mathcal{N}}\|_F^2 + \|\mathcal{W}\|_*\right\}$
14:     $\{\hat{w}_k(t)\}_{k=1}^{K} = \text{unfold}\left(\hat{\mathcal{W}}\right)$
15:     return $\{\hat{w}_k(t)\}_{k=1}^{K}$
16:   **else**
17:     $w'(t) = \frac{1}{K}\sum_{k \in S_t} w'_k(t)$
18:     return $w'(t)$
19:   **end if**
20: **end for**
21: return $w'(T)$

---

where $\mathcal{U} \in \mathbb{R}^{n_1 \times n_1 \times n_3}, \mathcal{V} \in \mathbb{R}^{n_2 \times n_2 \times n_3}$ are orthogonal, i.e., $\mathcal{U} * \mathcal{U}^H = \mathcal{I}$ and $\mathcal{V} * \mathcal{V}^H = \mathcal{I}$, where $(\cdot)^H$ denotes conjugate transpose. $\mathcal{S} \in \mathbb{R}^{n_1 \times n_2 \times n_3}$ is an $f$-diagonal tensor, whose each frontal slice is diagonal.

Note that tSVD can be efficiently computed in the Fourier domain using matrix SVD, as detailed in **Appendix** A.3. Furthermore, by truncating smaller singular values (or by retaining the larger singular values), we can decompose the tensor into a lower-rank part, referred to as the truncated tSVD (T-tSVD). We defer its algorithm to **Appendix** A.4.

## 3. Main Algorithm

### 3.1. FedCEO

Our main algorithm, as illustrated in Algorithm 2, represents a local version of the user-level differentially private federated learning framework, accompanied by a tensor low-rank proximal optimization acting on stacked noisy parameters

at the server. The specific objective function is as follows:

$$\hat{\mathcal{W}} = \arg\min_{\mathcal{W}} \left\{\lambda/\vartheta^{\frac{t}{I}} \|\mathcal{W} - \mathcal{W}_{\mathcal{N}}\|_F^2 + \|\mathcal{W}\|_*\right\}. \quad (2)$$

Every $I$ rounds, we fold the noisy parameters uploaded by clients into a third-order tensor $\mathcal{W}_{\mathcal{N}} \in \mathbb{R}^{d \times h \times K}$ where $d$ represents the input data dimension, $h$ denotes the network dimension, and $K$ signifies the number of clients selected in each round. Subsequently, we impose constraints on its low-rank structure using the previously introduced TNN, denoted as $\|\mathcal{W}\|_*$. Furthermore, to prevent trivial solutions, we apply an offset regularization term based on the Frobenius norm, denoted as $\|\mathcal{W} - \mathcal{W}_{\mathcal{N}}\|_F^2$. It also serves as a proximal operator, ensuring the convergence of the optimal point $\hat{\mathcal{W}}$ (Cai et al., 2010). In the next section, we prove that the constructed optimization objective in Eq. (2) is equivalent to the T-tSVD algorithm with the adaptive soft-thresholding rule in Theorem 3.1, where the truncation threshold is defined by a geometric series, denoted as $\frac{1}{2\lambda}\vartheta^{\frac{t}{I}}$.

### 3.2. Analysis

To elucidate the role of our low-rank proximal optimization objective at the server, we introduce the following theorem.

**Theorem 3.1** (Interpretability)**.** *For each* $\tau \geq 0$ *and* $\mathcal{W}_{\mathcal{N}} \in \mathbb{R}^{d \times h \times K}$, *our tensor low-rank proximal optimization objective defined in algorithm 2 obeys*

$$\text{T-tSVD}(\mathcal{W}_{\mathcal{N}}, \frac{1}{2\tau}) = \arg\min_{\mathcal{W}} \left\{\tau\|\mathcal{W} - \mathcal{W}_{\mathcal{N}}\|_F^2 + \|\mathcal{W}\|_*\right\},$$
$$(3)$$

*where* $\text{T-tSVD}(\cdot)$ *is a truncated tSVD operator and* $\frac{1}{2\tau}$ *is the truncation threshold, defined as follows:*

$$\text{T-tSVD}(\mathcal{W}, \frac{1}{2\tau}) := \mathcal{U} * \mathcal{D} * \mathcal{V}^H,$$

*where* $\mathcal{D}$ *is an f-diagonal tensor whose each frontal slice in the Fourier domain is* $\overline{\boldsymbol{D}}^{(i)}(j,j) = \max\{\overline{\boldsymbol{S}}^{(i)}(j,j) - \frac{1}{2\tau}, 0\}$, $j \leq \min(d,h), i = 1, \cdots, K$.

*Proof.* A proof is given in **Appendix** B.1     □

Combined with Remark B.2 deferred to **Appendix** B.1, it indicates that our approach contributes to a smoother global semantic space by flexibly truncating the high-frequency components of the parameter tensor.

Specifically, our theorem elucidates that as the regularization coefficient $\tau$ in Eq. (3) decreases, corresponding to a larger truncation threshold $\frac{1}{2\tau}$ in T-tSVD, leading to a smoother global semantic space. In our objective function, we set the coefficient to $\lambda/\vartheta^{\frac{t}{I}} (\vartheta > 1)$, corresponding to an adaptive threshold of $\frac{1}{2\lambda}\vartheta^{\frac{t}{I}}$. This implies that with the accumulation of noise (i.e., the increase in communication

rounds $t$), the server coordinates the gradual enhancement of semantic smoothness (*collaboration*) among the various clients, akin to a *CEO*. Furthermore, we can choose appropriate initialization coefficients $\lambda$ for different privacy settings. Specifically, for stronger privacy guarantees (larger Gaussian noise), selecting a smaller $\lambda$ results in a smoother global semantic space. Furthermore, we derive the following corollary.

**Proposition 3.2.** *Given that $\mathcal{W} \in \mathbb{R}^{d \times h \times K}$ and the regularization coefficient $\tau = \frac{1}{2\sigma_m}$ in Eq. (3), we have T-tSVD$(\mathcal{W}, \sigma_m)$ with truncation threshold $\sigma_m$. If $\sigma_m$ **larger than** the highest singular value of $\overline{W}^{(i)}$ for $i = 2, 3, \ldots, K$, the updated parameters in our **FedCEO** will degenerate to the low-rank approximation of the global parameter in **FedAvg**.*

*Formally, for all $k = 1, \ldots, K$ we have*

$$[\text{T-tSVD}(\mathcal{W}, \sigma_m)]^{(k)} = \text{TruncatedSVD}(W, \frac{\sigma_m}{K})$$
$$\approx W\left(\sigma_m = \max_{2 \le i \le n}\left\{\sigma_r(\overline{W}^{(i)})\right\}\right),$$

*where $\mathcal{W}$ is the parameter tensor in FedCEO, $W$ is the aggerated average parameter in FedAvg and $\text{TruncatedSVD}(\cdot)$ is the truncated SVD operator for matrices (Defined in **Appendix A.4**).*

*Proof.* A proof is given in **Appendix B.2**. □

Hence, we can infer that when the truncation threshold is appropriately chosen (allowing the parameter tensor to retain only the first frequency component), our method can approximately degenerate into FedAvg. Based on the above analysis, we can visualize the low-rank proximal optimization process through dynamic singular value truncation in the spectral space, as illustrated in Figure 2.

**FedAvg vs. FedCEO**: In contrast to our FedCEO, FedAvg implies retaining only the lowest-frequency components in the spectral space, representing coarse-grained *mutual collaboration* that lacks the adaptability to different DP settings and continuous training processes. Our approach, characterized by flexible complementarity of semantic information due to adaptive truncation thresholds, demonstrates superior model utility in DPFL.

### 3.3. On the Scalability

The complexity of our approach is $O(\sum_{l=1}^{L} K \cdot \min(d_l d_{l-1}^2, d_l^2 d_{l-1}))$ where $L$ is the number of layers, $K$ is the number of clients and $\min(d_l d_{l-1}^2, d_l^2 d_{l-1})$ is the complexity of matrix-SVD for each layers. To scale to larger models, we can perform T-tSVD to the last few layers only since these layers will be narrower than other layers but contain more semantic information.

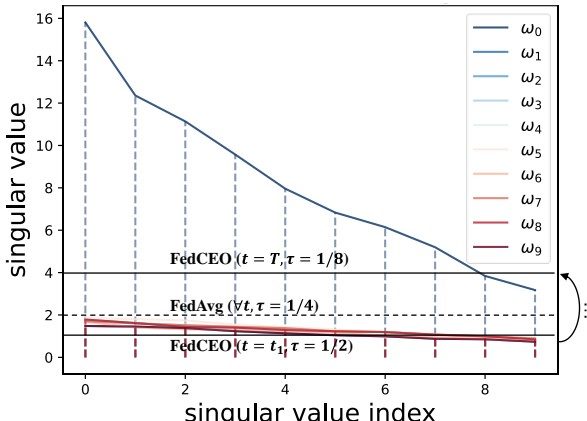

*Figure 2.* The visualization of the low-rank proximal optimization process, where $w_i$ is the $(i+1)$-th frequency component.

## 4. Utility-Privacy Trade-off Analysis

In this section, we provide a theoretical analysis of the utility and privacy guarantees for our FedCEO. We denote $\epsilon_u$ and $\epsilon_p$ as the model utility and the privacy budget, respectively. Furthermore, we establish an improved utility-privacy trade-off bound $\epsilon_u \cdot \epsilon_p \le O(\sqrt{d}/K)$. In comparison to recent advanced work, our approach exhibits an improvement of $O(\sqrt{d})$ over the current SOTA result (i.e. $O(d/K)$ as reported in (Shen et al., 2023)).

### 4.1. Utility Analysis

Firstly, we introduce a generalized definition of utility, which encompasses the definitions in the latest works.

**Definition 4.1** (Utility Loss (Zhang et al., 2022))**.** The model utility is characterized as the difference in performance when utilizing the protected model information sampled from the distribution $P'_k$ compared to that drawn from the unprotected distribution $P_k$,

$$\epsilon := \frac{1}{K}\sum_{k=1}^{K}\epsilon_k = \frac{1}{K}\sum_{k=1}^{K}\left[U_k(P'_k) - U_k(P_k)\right],$$

where $P'_k$ and $P_k$ represent, respectively, the distributions of *convergent models* with or without DP. $U_k(P) = \mathbb{E}_{D_k}\mathbb{E}_{W_k}\left[\frac{1}{B}\sum_{b\in\mathcal{B}}U(W_k, b)\right]$ denotes the expected utility taken with respect to $W_k \sim P'_k$ or $P_k$ and $U$ is any metric measuring the performance of the model.

In this paper, we denote $P_k$ as the local model distribution in FedAvg without DP, and $P'_k$ as the local model distribution in our FedCEO with DP. Let $U$ represent the empirical loss of the model. We define our utility as follows by setting $P_k = w_k$, $P'_k = w'_k$, and $U = F$.

**Definition 4.2** (Model Utility). We define the utility of the model within the differential privacy federated learning framework as follows:

$$\epsilon_u := \frac{1}{K}\sum_{k=1}^{K}\epsilon_{u,k} = \frac{1}{K}\sum_{k=1}^{K}\left[f_k(w'_k) - f_k(w_k)\right]$$
$$= f(w') - f(w^*),$$

where $w'$ and $w^*$ represent, respectively, *convergent global models* of FedCEO or FedAvg. $f_k$ denotes local empirical risk and $f$ denotes global empirical risk.

Based on the above definitions, we present the utility guarantee for our FedCEO as follows.

**Theorem 4.3** (Utility Analysis of FedCEO). *Suppose that Assumptions B.5, B.6 and B.7 hold and let local empirical risk $f_k$ satisfies Definition B.4. Set $\tau = \frac{1}{2(\tau_0/K)}$ in Eq. (3) and $0 < m < 1/\mu$, where $m = \eta - \frac{L_2 B^2(1+\gamma)^2}{2\mu^2}$*
$$-\left[\frac{\sqrt{2}\left(\mu B(1+\gamma)+2L_2(1+\gamma)^2 B^2\right)}{\sqrt{K}\mu^2} + \frac{2L_2(1+\gamma)^2 B^2}{K\mu^2}\right].$$

*Algorithm 2 satisfies*

$$\epsilon_u = f(w') - f(w^*) \leq \frac{\sqrt{2}L_1}{K}\left(\sqrt{r}\tau_0 + \sqrt{d}\sqrt{h}C\sigma\right)\frac{1}{2\mu m},$$

*where $r$ is the rank of the parameter tensor after our processing and $d$ is the dimension of input data.*

*Proof.* A proof is given in **Appendix** B.3 ☐

In summary, we demonstrate that our FedCEO satisfies an outstanding utility bound with $\epsilon_u \leq O(\frac{\sqrt{r}+\sqrt{d}}{K})$, providing theoretical utility guarantees. Moreover, achieving low-rank global semantic space of the whole parameters (i.e. $r \ll d$), we have $\epsilon_u \leq O(\frac{\sqrt{d}}{K})$.

## 4.2. Privacy Analysis

We first establish a general privacy guarantee for Algorithm 1, extending the theoretical results of data-level DP from (Abadi et al., 2016) to user-level DP.

**Lemma 4.4** (Privacy Analysis of UDP-FedAvg (Cheng et al., 2022)). *There exist constants $c_1$ and $c_2$ so that given the clients sampling probability $p = \frac{K}{N}$ and the number of communication rounds $T$, for any $\epsilon < c_1 p^2 T$, Algorithm 1 satisfies user-level $(\epsilon, \delta)$-DP for any $\delta > 0$ if we choose $\sigma \geq \frac{c_2 p\sqrt{T\log(1/\delta)}}{\epsilon}$.*

The above lemma proves that the model parameters uploaded by each client have strict privacy guarantees. Furthermore, taking into account the tensor low-rank proximal optimization at the server, we present the privacy guarantee for our FedCEO as follows.

**Theorem 4.5** (Privacy Analysis of FedCEO). *Suppose that the privacy budget $\epsilon_p < c_1 q^2 T$, let $\sigma$ be the noise multiple (a tunable parameter that controls the privacy-utility trade-off), Algorithm 2 satisfies user-level $(\epsilon_p, \delta)$-DP with*

$$\epsilon_p = \frac{c_2 K\sqrt{T\log(1/\delta)}}{N\sigma}.$$

*Proof.* A proof is given in **Appendix** B.4 ☐

In summary, we demonstrate that our FedCEO satisfies $(\epsilon_p, \delta)$-differential privacy with privacy budget $\epsilon_p = \frac{c_2 K\sqrt{T\log(1/\delta)}}{N\sigma}$, providing theoretical privacy guarantees.

## 4.3. Guaranteed Improvement of Utility-Privacy Trade-off

On the one hand, Theorem 4.3 indicates that achieving high utility (i.e., a small $\epsilon_u$) necessitates selecting a *small* noise multiplier $\sigma$. On the other hand, Theorem 4.5 asserts that achieving high privacy (i.e., a small $\epsilon_p$) necessitates selecting a *large* noise multiplier $\sigma$. Next, we unify the utility and privacy analyses of our FedCEO in the DPFL setting to establish the overarching utility-privacy trade-off.

**Corollary 4.6.** *Let $\epsilon_u$ and $\epsilon_p$ denote the model utility and the privacy budget, respectively. Under the Assumptions B.5 to B.7 and the conditions outlined in Theorems 4.3 and 4.5, Algorithm 2 satisfies*

$$\epsilon_u \cdot \epsilon_p \leq \frac{\hat{c}\sqrt{d}}{N},$$

*where $d$ is the input dimension and $N$ is the total number of clients. $\hat{c}$ hides the constants and $\log$ terms.*

Overall, our FedCEO achieves a utility-privacy trade-off bound $\epsilon_u \cdot \epsilon_p \leq O\left(\frac{\sqrt{d}}{N}\right)$.

**Summary**: Under the unified settings of utility (Definition 4.1) and privacy (Definition 2.1), compared to existing SOTA results, we improve the utility-privacy trade-off bound from $O(d^{1.5}/N)$ (Theorem 5.2 in (Jain et al., 2021)) and $O(d/N)$ (Corollary 5.1 in CENTAUR (Shen et al., 2023)) to $O(\sqrt{d}/N)$. In contrast, **our approach leverages higher-order tensor algorithms to integrate the processes of parameter partition and semantic fusion, resulting in a notable improvement with $O(\sqrt{d})$ and providing a better guarantee for the trade-off between utility and privacy in DPFL.**

## 5. Experiments

In this section, we present empirical results on real-world federated learning datasets, validating the utility guarantees, privacy guarantees, and the advanced trade-off between the two provided by our FedCEO algorithm.

*Table 1.* Testing accuracy (%) on EMNIST and CIFAR-10 under $\delta = 10^{-5}$ and various privacy settings with three common $\sigma_g$. A larger $\sigma_g$ indicates a smaller $\epsilon_p$, i.e. the stronger privacy guarantee. $\vartheta$ stands for the scaling factor of the parameter $\lambda$.

| Dataset | Model | Setting | UDP-FedAvg | PPSGD | CENTAUR | FedCEO ($\vartheta=1$) | FedCEO ($\vartheta > 1$) |
|---------|-------|---------|------------|-------|---------|-----------------------|-------------------------|
| EMNIST | MLP-2-Layers | $\sigma_g = 1.0$ | 76.59% | 77.01% | 77.26% | 77.14% | **78.05%** |
| | | $\sigma_g = 1.5$ | 69.91% | 70.78% | 71.86% | 71.56% | **72.44%** |
| | | $\sigma_g = 2.0$ | 60.32% | 61.51% | 62.12% | 63.38% | **64.20%** |
| CIFAR-10 | LeNet-5 | $\sigma_g = 1.0$ | 43.87% | 49.24% | 50.14% | 50.09% | **54.16%** |
| | | $\sigma_g = 1.5$ | 34.34% | 47.56% | 46.90% | 48.89% | **50.00%** |
| | | $\sigma_g = 2.0$ | 26.88% | 34.61% | 36.70% | 37.39% | **45.35%** |



(a) Privacy Attack on FedAvg (PSNR=19.73)  (b) Privacy Attack on UDP-FedAvg (PSNR=10.33)  (c) Privacy Attack on FedCEO (PSNR=10.15)

*Figure 3.* Privacy protection performance of three federated learning frameworks on CIFAR-10. Both FedCEO and UDP-FedAvg demonstrate robust defense against privacy attacks with smaller *Peak Signal-to-Noise Ratio* (PSNR), while DLG successfully infers sensitive images from clients in FedAvg.

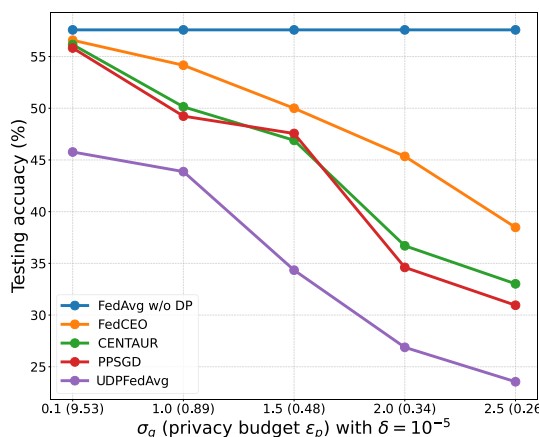

*Figure 4.* Utility-Privacy Trade-off for our FedCEO and other methods on CIFAR-10.

## 5.1. Experiment Setup

We set the clipping threshold $C$ uniformly to 1.0 and use three different privacy settings corresponding to $\sigma_g = 1.0, 1.5$, and 2.0. $\sigma_g$ is proportional to the noise multiplier $\sigma$, defined in the privacy engine from the Opacus package (Yousefpour et al., 2021). We conduct experimental evaluations on both MLP and LeNet model architectures, reporting the testing accuracy of the global model (consistent with representative papers (Bietti et al., 2022; Shen et al., 2023) from recent years). Specific model structures and hyperparameters are detailed in **Appendix** C.1.

## 5.2. Utility Experiments

In Table 1, our FedCEO demonstrates state-of-the-art performance under different privacy settings, aligning with the utility analysis in Section 4.1. Additionally, we can adapt to various privacy settings by flexibly adjusting the initial coefficient $\lambda$. Specifically, larger $\sigma_g$ corresponds to smaller $\lambda$ (lower rank, i.e. smaller $r$). Moreover, our adaptive mechanism increases the truncation threshold (i.e. $\frac{1}{2\lambda} \vartheta^{\frac{t}{T}}$) as the Gaussian noise accumulates during the FL training process, due to the geometric series with a common ratio $\vartheta > 1$. Refer to **Appendix** C.2.1, C.2.2 and C.2.3 for experiments about model training efficiency, heterogeneous FL settings and other local model as well text dataset.

## 5.3. Privacy Experiments

In Figure 3, our FedCEO and UDP-FedAvg exhibit similar privacy protection performance for user data, validating the privacy analysis in Section 4.2. Specifically, we input the locally uploaded LeNet model (gradients) from three different FL frameworks into the *Deep Leakage from Gradients* (DLG) (Zhu et al., 2019) algorithm and set attack iterations to 300 with a learning rate of 0.01. For our FedCEO, the DLG attack is conducted after the tensor low-rank proximal optimization to verify that our operation does not compromise the privacy protection of DP. Then we report the inference images and their PSNR (dB, ↓) values in Figure 3. Refer to **Appendix** C.2.4 for more details.

## 5.4. Improved Utility-Privacy Trade-off

In Figure 4, we observe that our FedCEO exhibits the best utility performance across various privacy settings. Moreover, compared to other privacy-preserving methods, our model shows the smallest decrease in testing accuracy as privacy is enhanced (i.e. $\sigma_g$ increases and $\epsilon_p$ decreases). This indicates that our FedCEO achieves a best utility-privacy trade-off, consistent with the analysis in Section 4.3.

# 6. Conclusion

In this paper, we explore the concept of *flexible semantic smoothness (collaboration)* among clients, providing theoretical guarantees for both utility and privacy. Our approach achieves an improved utility-privacy trade-off bound of $O(\sqrt{d})$ compared to the current SOTA results. Furthermore, we conduct comprehensive experiments to validate the utility and privacy across different datasets and privacy settings, demonstrating the advanced performance consistent with our theoretical analysis. In the future, we aim to extend tensor low-rank techniques to heterogeneous federated learning, addressing challenges of gradient conflicts.

# Impact Statement

This paper presents work whose goal is to advance the field of Machine Learning. There are many potential societal consequences of our work, none of which we feel must be specifically highlighted here.

# Acknowledgements

The research is supported by the National Key R&D Program of China (2023YFB2703700), the National Natural Science Foundation of China (62176269), the Natural Science Foundation of Guangdong (2023A1515012026).

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

# A. More on Preliminaries

## A.1. Discrete Fourier Transform

For each vector $\boldsymbol{v} \in \mathbb{R}^n$, it defines the discrete Fourier transform, denoted as $\mathrm{DFT}(\cdot)$, as follows:

$$\overline{\boldsymbol{v}} = \mathrm{DFT}(\boldsymbol{v}) := \boldsymbol{F}_n \boldsymbol{v} \in \mathbb{C}^n, (\boldsymbol{F}_n)_{jk} = \left[ \omega^{(j-1)(k-1)} \right] \tag{4}$$

Here, $\boldsymbol{F}_n \in \mathbb{C}^{n \times n}$ is the DFT matrix and $\omega = \mathrm{e}^{-i\frac{2\pi}{n}}$ where $e$ represents the base of the natural logarithm, and $i$ denotes the imaginary unit. Furthermore, we denote the inverse discrete Fourier transform as $\mathrm{IDFT}(\cdot)$.

## A.2. T-product

For each third-order tensor $\mathcal{U} \in \mathbb{R}^{n_1 \times n_2 \times n_3}$ and $\mathcal{V} \in \mathbb{R}^{n_2 \times n_4 \times n_3}$, it defines the t-product, denoted as $\mathcal{U} * \mathcal{V}$, as follows:

$$\mathcal{U} * \mathcal{V} := \mathrm{fold}(\mathrm{bcirc}(\mathcal{U}) \cdot \mathrm{unfold}(\mathcal{V})) \in \mathbb{R}^{n_1 \times n_4 \times n_3}, \tag{5}$$

where the $\mathrm{bcirc}(\cdot)$ is a tensor matricization operator known as block circulant matricization, denoted by

$$\mathrm{bcirc}(\mathcal{U}) := \begin{bmatrix} \boldsymbol{U}^{(1)} & \boldsymbol{U}^{(n_3)} & \cdots & \boldsymbol{U}^{(2)} \\ \boldsymbol{U}^{(2)} & \boldsymbol{U}^{(1)} & \cdots & \boldsymbol{U}^{(3)} \\ \vdots & \vdots & \ddots & \vdots \\ \boldsymbol{U}^{(n_3)} & \boldsymbol{U}^{(n_3-1)} & \cdots & \boldsymbol{U}^{(1)} \end{bmatrix} \in \mathbb{R}^{n_1 n_3 \times n_2 n_3}.$$

Furthermore, we define the tensor unfolding operator as follows:

$$\mathrm{unfold}(\mathcal{V}) := \begin{bmatrix} \boldsymbol{V}^{(1)} \\ \boldsymbol{V}^{(2)} \\ \vdots \\ \boldsymbol{V}^{(n_3)} \end{bmatrix} \in \mathbb{R}^{n_2 n_3 \times n_4}, \quad \mathrm{fold}(\mathrm{unfold}(\mathcal{V})) = \mathcal{V} \in \mathbb{R}^{n_2 \times n_4 \times n_3},$$

where $\boldsymbol{V}^{(i)}$ represents the $i$-th frontal slice of tensor $\mathcal{V}$, i.e., $\mathcal{V}(:,:,i)$.

## A.3. Algorithm of tSVD

For each matrix $Y \in \mathbb{R}^{n_1 \times n_2}$ of rank $r$, it defines the singular value decomposition operator, denoted as $\mathrm{SVD}(\cdot)$, as follows:

$$\mathrm{SVD}(\boldsymbol{Y}) := \boldsymbol{U}\boldsymbol{\Sigma}\boldsymbol{V}^T, \quad \boldsymbol{\Sigma} = \mathrm{diag}\left(\{\sigma_i\}_{i=1}^r\right), \tag{6}$$

where $\boldsymbol{U} \in \mathbb{R}^{n_1 \times n_1}, \boldsymbol{V} \in \mathbb{R}^{n_2 \times n_2}$ are orthogonal and $\boldsymbol{\Sigma} \in \mathbb{R}^{n_1 \times n_2}$ is a diagonal matrix, whose main diagonal elements are the singular values of the matrix $Y$. Next, we introduce the algorithm of singular value decomposition for third-order tensors.

---

**Algorithm 3** Tensor singular value decomposition (tSVD)

  **Input:** $\mathcal{W} \in \mathbb{R}^{n_1 \times n_2 \times n_3}$
  **Output:** $\mathcal{U} \in \mathbb{R}^{n_1 \times n_1 \times n_3}, \mathcal{S} \in \mathbb{R}^{n_1 \times n_2 \times n_3}, \mathcal{V} \in \mathbb{R}^{n_2 \times n_2 \times n_3}$
1:  $\overline{\mathcal{W}} = \mathrm{DFT}(\mathcal{W}, 3)$
2: **for** $i = 1$ to $n_3$ **do**
3:    $[\overline{\boldsymbol{U}}^{(i)}, \overline{\boldsymbol{S}}^{(i)}, \overline{\boldsymbol{V}}^{(i)}] = \mathrm{SVD}(\overline{\boldsymbol{W}}^{(i)})$
4: **end for**
5:  $\overline{\mathcal{U}} = \mathrm{fold}\left(\left[\overline{\boldsymbol{U}}^{(1)}, \cdots, \overline{\boldsymbol{U}}^{(n_3)}\right]^T\right), \overline{\mathcal{S}} = \mathrm{fold}\left(\left[\overline{\boldsymbol{S}}^{(1)}, \cdots, \overline{\boldsymbol{S}}^{(n_3)}\right]^T\right), \overline{\mathcal{V}} = \mathrm{fold}\left(\left[\overline{\boldsymbol{V}}^{(1)}, \cdots, \overline{\boldsymbol{V}}^{(n_3)}\right]^T\right)$
6:  $\mathcal{U} = \mathrm{IDFT}(\overline{\mathcal{U}}, 3), \mathcal{S} = \mathrm{IDFT}(\overline{\mathcal{S}}, 3), \mathcal{V} = \mathrm{IDFT}(\overline{\mathcal{V}}, 3)$

---

## A.4. Algorithm of T-tSVD

For each matrix $Y \in \mathbb{R}^{n_1 \times n_2}$ of rank $r$, it defines the truncated singular value decomposition operator, denoted as TruncatedSVD($\cdot$), as follows:

$$\text{TruncatedSVD}(\boldsymbol{Y}, \tau) := \boldsymbol{U} \mathcal{D}_\tau (\boldsymbol{\Sigma}) \boldsymbol{V}^T, \quad \mathcal{D}_\tau (\boldsymbol{\Sigma}) = \text{diag}\left(\{\max(\sigma_i - \tau, 0)\}_{i=1}^r\right), \tag{7}$$

where $\tau$ is the truncated threshold. Next, we introduce the algorithm of truncated singular value decomposition for third-order tensors.

---

**Algorithm 4** Truncated tensor singular value decomposition (T-tSVD)

---

**Input:** $\mathcal{W} \in \mathbb{R}^{n_1 \times n_2 \times n_3}$
**Output:** $\mathcal{U} \in \mathbb{R}^{n_1 \times n_1 \times n_3}, \mathcal{D} \in \mathbb{R}^{n_1 \times n_2 \times n_3}, \mathcal{V} \in \mathbb{R}^{n_2 \times n_2 \times n_3}$
1: $\overline{\mathcal{W}} = \text{DFT}(\mathcal{W}, 3)$
2: **for** $i = 1$ to $n_3$ **do**
3: $\quad [\overline{\boldsymbol{U}}^{(i)}, \overline{\boldsymbol{D}}^{(i)}, \overline{\boldsymbol{V}}^{(i)}] = \text{TruncatedSVD}(\overline{\boldsymbol{W}}^{(i)}, \tau)$
4: **end for**
5: $\overline{\mathcal{U}} = \text{fold}\left(\left[\overline{\boldsymbol{U}}^{(1)}, \cdots, \overline{\boldsymbol{U}}^{(n_3)}\right]^T\right), \overline{\mathcal{D}} = \text{fold}\left(\left[\overline{\boldsymbol{D}}^{(1)}, \cdots, \overline{\boldsymbol{D}}^{(n_3)}\right]^T\right), \overline{\mathcal{V}} = \text{fold}\left(\left[\overline{\boldsymbol{V}}^{(1)}, \cdots, \overline{\boldsymbol{V}}^{(n_3)}\right]^T\right)$
6: $\mathcal{U} = \text{IDFT}(\overline{\mathcal{U}}, 3), \mathcal{D} = \text{IDFT}(\overline{\mathcal{D}}, 3), \mathcal{V} = \text{IDFT}(\overline{\mathcal{V}}, 3)$

---

# B. Proofs of Main Theory

## B.1. Proof of Theorem 3.1 (Interpretability of our optimization objective)

*Proof.* Based on Parseval's theorem and the definition of the tensor nuclear norm (Definition 2.2), we have

$$\tau \|\mathcal{W} - \mathcal{W}_{\mathcal{N}}\|_F^2 + \|\mathcal{W}\|_*$$

$$= \frac{\tau}{K} \|\text{DFT}(\mathcal{W} - \mathcal{W}_{\mathcal{N}})\|_F^2 + \frac{1}{K} \sum_{i=1}^K \|\overline{\boldsymbol{W}}^{(i)}\|_*$$

$$= \frac{\tau}{K} \sum_{i=1}^K \|\overline{\boldsymbol{W}}^{(i)} - \overline{\boldsymbol{W}}_{\mathcal{N}}^{(i)}\|_F^2 + \frac{1}{K} \sum_{i=1}^K \|\overline{\boldsymbol{W}}^{(i)}\|_*, \tag{8}$$

where $\overline{\boldsymbol{W}}^{(i)}$ represents the $i$-th frontal slice of the tensor obtained by applying the DFT along the third dimension on $\mathcal{W}$. Therefore, we know

$$\min_{\mathcal{W}} \left\{ \tau \|\mathcal{W} - \mathcal{W}_{\mathcal{N}}\|_F^2 + \|\mathcal{W}\|_* \right\} \Leftrightarrow \left\{ \min_{\overline{\boldsymbol{W}}^{(i)}} \left( \tau \|\overline{\boldsymbol{W}}^{(i)} - \overline{\boldsymbol{W}}_{\mathcal{N}}^{(i)}\|_F^2 + \|\overline{\boldsymbol{W}}^{(i)}\|_* \right) \right\}_{i=1}^K. \tag{9}$$

By Lemma B.1, we have

$$\text{TruncatedSVD}(\overline{\boldsymbol{W}}_{\mathcal{N}}^{(i)}, \frac{1}{2\tau}) = \arg\min_{\overline{\boldsymbol{W}}^{(i)}} \left\{ \tau \|\overline{\boldsymbol{W}}^{(i)} - \overline{\boldsymbol{W}}_{\mathcal{N}}^{(i)}\|_F^2 + \|\overline{\boldsymbol{W}}^{(i)}\|_* \right\}. \tag{10}$$

Now, let us define $\hat{\mathcal{W}} = \arg\min_{\mathcal{W}} \left\{ \tau \|\mathcal{W} - \mathcal{W}_{\mathcal{N}}\|_F^2 + \|\mathcal{W}\|_* \right\}$. Then, based on Eq. (9), Eq. (10) and algorithm A.4, we have

$$\hat{\mathcal{W}} = \text{IDFT} \left\{ \text{fold} \left( \left[ \text{TruncatedSVD}(\overline{\boldsymbol{W}}_{\mathcal{N}}^{(1)}, \frac{1}{2\tau}), \cdots, \text{TruncatedSVD}(\overline{\boldsymbol{W}}_{\mathcal{N}}^{(K)}, \frac{1}{2\tau}) \right]^T \right) \right\}$$

$$= \text{T-tSVD}(\mathcal{W}_{\mathcal{N}}, \frac{1}{2\tau}) \tag{11}$$

$\square$

**Lemma B.1** (SVT Lemma (Cai et al., 2010)). *For each $\tau \geq 0$ and $\boldsymbol{Y} \in \mathbb{R}^{n_1 \times n_2}$, the truncated singular value decomposition operator* $\mathrm{TruncatedSVD}(\cdot)$ *obeys*

$$\mathrm{TruncatedSVD}(\boldsymbol{Y}, \frac{1}{2\tau}) = \arg\min_{\boldsymbol{X}} \left\{ \tau \|\boldsymbol{X} - \boldsymbol{Y}\|_F^2 + \|\boldsymbol{X}\|_* \right\}.$$

*Remark* B.2. Due to the inherent correlation within the semantic space of clients, if we perform the Fourier transform on parameter tensor along the third dimension, the singular values of its first slice will be much greater than the later ones (see Fig. 5). Formally, we can express the assumption as follows,

$$\sigma_r\big(\overline{\boldsymbol{W}}^{(1)}\big) \gg \sigma_r\big(\overline{\boldsymbol{W}}^{(i)}\big)$$

where $\sigma_r(\cdot)$ is the $r$-th highest singular value of a matrix and $i = 2, 3, \ldots, K$.

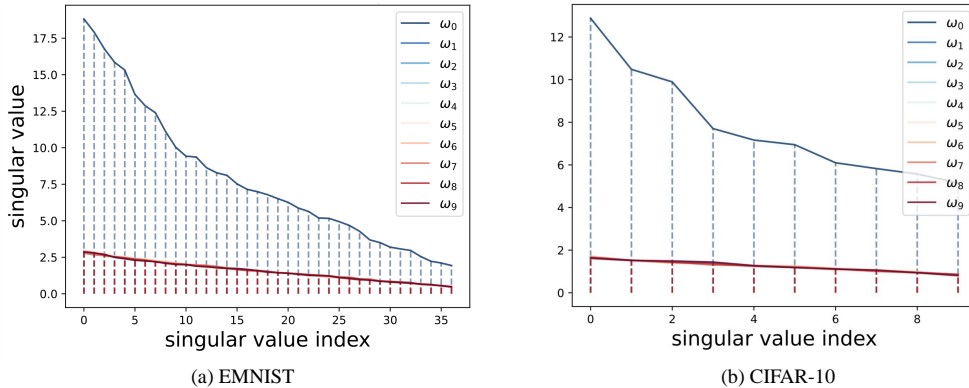

(a) EMNIST           (b) CIFAR-10

*Figure 5.* The singular value curves in spectral space for the original noise parameter tensors on two real datasets. $\omega_0$ to $\omega_9$ represent components from low to high frequency.

### B.2. Proof of Proposition 3.2

Next, we have the following proof,

*Proof.* First, consider T-tSVD($\mathcal{W}, \sigma_m$) in Fourier space and we have

$$\overline{\boldsymbol{D}}^{(i)}(j, j) = \max\{\overline{\boldsymbol{S}}^{(i)}(j, j) - \sigma_m, 0\}. \tag{12}$$

Since we set $\sigma_m$ to be **larger than** the highest singular value of $\overline{\boldsymbol{W}}^{(i)}$ for $i = 2, 3, \ldots, K$, we have

$$\overline{\boldsymbol{D}}^{(i)}(j, j) = 0 \tag{13}$$

for all $i = 2, 3, \ldots, K$ and

$$\overline{\boldsymbol{D}}^{(1)}(j, j) = \max\{\overline{\boldsymbol{S}}^{(1)}(j, j) - \sigma_m, 0\}. \tag{14}$$

It has at least one value to be not zero by Remark B.2.

Thus, when we perform the inverse Fourier transform, we will get

$$
\begin{aligned}
[\text{T-tSVD}(\mathcal{W}, \sigma_m)]^{(k)} &= \frac{1}{K} \sum_{j=1}^{K} e^{i\frac{2\pi}{K}(j-1)(k-1)} \overline{U}^{(j)} \overline{D}^{(j)} \overline{V}^{(j)} \\
&= \frac{1}{K} e^{i\frac{2\pi}{K}(k-1)\cdot 0} \overline{U}^{(1)} \overline{D}^{(1)} \overline{V}^{(1)} \\
&= \frac{1}{K} \overline{U}^{(1)} \overline{D}^{(1)} \overline{V}^{(1)} \\
&= \frac{1}{K} \text{TruncatedSVD}(\overline{W}^{(1)}, \sigma_m).
\end{aligned}
\tag{15}
$$

From the other side, we know that

$$
\overline{W}^{(1)} = \sum_{k=1}^{K} e^{-i\frac{2\pi}{K}0\cdot(k-1)} W^{(k)} = \sum_{k=1}^{K} W^{(k)} = KW.
\tag{16}
$$

Therefore we obtain that

$$
[\text{T-tSVD}(\mathcal{W}, \sigma_m)]^{(k)} = \frac{1}{K} \text{TruncatedSVD}(KW, \sigma_m) = \text{TruncatedSVD}(W, \frac{\sigma_m}{K}).
\tag{17}
$$

Moreover, we appropriately set $\sigma_m = \max_{2 \leq i \leq n} \left\{ \sigma_r(\overline{W}^{(i)}) \right\}$.

On the one hand, normally we know that $K \gg \sigma_m$. So, based on Eq. (17), we have

$$
\begin{aligned}
[\text{T-tSVD}(\mathcal{W}, \sigma_m)]^{(k)} &= \text{TruncatedSVD}(W, \frac{\sigma_m}{K}) \\
&\approx W.
\end{aligned}
\tag{18}
$$

On the other hand, by Remark B.2, Eq. (15) and Eq. (16), we have

$$
\begin{aligned}
[\text{T-tSVD}(\mathcal{W}, \sigma_m)]^{(k)} &= \frac{1}{K} \text{TruncatedSVD}(\overline{W}^{(1)}, \sigma_m) \\
&\approx \frac{1}{K} \overline{W}^{(1)} = W.
\end{aligned}
\tag{19}
$$

$\square$

### B.3. Proof of Theorem 4.3 (Utility Analysis of Our FedCEO)

Let $w'(t)$ denote the global model at $t$-th round in Algorithm 2. Let $w_k(t+1)$ denote the inexact solution (defined in B.3) of local optimization without DP by $\min_w f_k(w; w'(t))$ at $t+1$ round, where $w'(t)$ is the initial solution. Let $\mathcal{M}$ denote the DP mechanism we used. Let $\mathcal{F}$ denote our low-rank proximal optimization at the server (i.e. adaptive T-tSVD). Then we have $\hat{w}_k(t+1) = \mathcal{F}\left(\{\mathcal{M}(w_k(t+1))\}_{k=1}^{K}\right)$ and define $w'(t+1) = \mathbb{E}_{S_t}[\hat{w}_k(t+1)]$. Moreover, we make the following definitions and assumptions.

**Definition B.3** ($\gamma_k^{t+1}$-inexact solution). For a function $f_k(w)$ and $\gamma_k^{t+1} \in [0,1]$, we define $w_k(t+1)$ as a $\gamma_k^{t+1}$-inexact solution of $\min_w f_k(w; w_0)$ at $(t+1)$ round if

$$
\|\nabla f_k(w_k(t+1))\| \leq \gamma_k^t \|\nabla f_k(w_0)\|,
$$

where $w_0$ is the initial solution in optimization. $\gamma_k^{t+1}$ represents the local computational load of the $k$-th client at $(t+1)$-th round.

**Definition B.4** ($B$-local dissimilarity). The local empirical risk $f_k$ are $B$-locally dissimilar at $w$ if

$$
\mathbb{E}_k\left[\|\nabla f_k(w)\|^2\right] \leq \|\nabla f(w)\|^2 B^2.
$$

Moreover, we define $B(w) = \sqrt{\frac{\mathbb{E}_k[\|\nabla f_k(w)\|^2]}{\|\nabla f(w)\|^2}}$ for $\|\nabla f(w)\| \neq 0$ and then $B(w) \leq B$.

**Assumption B.5** ($L_1$-continuous). $f_1, \cdots, f_N$ are all $L_1$-continuous:

$$\forall \boldsymbol{v}, \boldsymbol{w}, \quad \|f_k(\boldsymbol{v}) - f_k(\boldsymbol{w})\| \leq L_1 \|\boldsymbol{v} - \boldsymbol{w}\|.$$

**Assumption B.6** ($L_2$-smooth). $f_1, \cdots, f_N$ are all $L_2$-smooth:

$$\forall \boldsymbol{v}, \boldsymbol{w}, \quad \|\nabla f_k(\boldsymbol{v}) - \nabla f_k(\boldsymbol{w})\| \leq L_2 \|\boldsymbol{v} - \boldsymbol{w}\|.$$

**Assumption B.7** ($\mu$-strongly convex). $f_1, \cdots, f_N$ are all $\mu$-strongly convex:

$$\forall \boldsymbol{v}, \boldsymbol{w}, \quad (\nabla f_k(\boldsymbol{v}) - \nabla f_k(\boldsymbol{w}))^T (\boldsymbol{v} - \boldsymbol{w}) \geq \mu \|\boldsymbol{v} - \boldsymbol{w}\|^2.$$

We can also study the utility guarantee of our FedCEO algorithm under more general non-convex objectives if we introduce the proximal term to $f_k$ like FedProx (Li et al., 2020).

*Proof.* At $(t+1)$-th round, considering **local optimization without DP**:

Let us define $w_k^*(t+1) = \arg\min_w f_k(w; w'(t))$ (i.e. exact optimal solution). So, we know

$$\nabla f_k [w_k^*(t+1)] = 0. \tag{20}$$

Then, based on Assumption B.7, we have

$$(\nabla f_k(\boldsymbol{v}) - \nabla f_k(\boldsymbol{w}))^T (\boldsymbol{v} - \boldsymbol{w}) \geq \mu \|\boldsymbol{v} - \boldsymbol{w}\|^2. \tag{21}$$

Set $\boldsymbol{v} = w_k^*(t+1), \boldsymbol{w} = w_k(t+1)$, based on Eq. (20) and Definition B.3, we have

$$\|w_k^*(t+1) - w_k(t+1)\| \leq \frac{1}{\mu} \|\nabla f_k(w_k(t+1))\| \leq \frac{\gamma_k^{t+1}}{\mu} \|\nabla f_k(w'(t))\|. \tag{22}$$

Similarly, we know that

$$\|w_k^*(t+1) - w'(t)\| \leq \frac{1}{\mu} \|\nabla f_k(w'(t))\|. \tag{23}$$

Combining Eq. (22) and Eq. (23), based on the triangle inequality, we have

$$\|w_k(t+1) - w'(t)\| \leq \|w_k^*(t+1) - w_k(t+1)\| + \|w_k^*(t+1) - w'(t)\| \leq \frac{1 + \gamma_k^{t+1}}{\mu} \|\nabla f_k(w'(t))\|. \tag{24}$$

Next, let us define $\bar{w}(t+1) = \mathbb{E}_k[w_k(t+1)]$ (an auxiliary variable). Based on the Jensen's inequality and Definition B.4, we have

$$\begin{aligned}
\|\bar{w}(t+1) - w'(t)\| \leq \mathbb{E}_k[\|w_k(t+1) - w'(t)\|] &\leq \frac{1 + \gamma_k^{t+1}}{\mu} \mathbb{E}_k[\|\nabla f_k(w'(t))\|] \\
&\leq \frac{1 + \gamma_k^{t+1}}{\mu} \sqrt{\mathbb{E}_k\left[\|\nabla f_k(w'(t))\|^2\right]} \\
&\leq \frac{(1 + \gamma_k^{t+1})}{\mu} B \|\nabla f(w'(t))\|.
\end{aligned} \tag{25}$$

According to the local updates from gradient descent, we have

$$w_k(t+1) = w'(t) - \eta \nabla f_k(w'(t)). \tag{26}$$

Based on the Taylor expansion and Assumption B.6, we have

$$\begin{aligned}
f(\bar{w}(t+1)) &\leq f(w'(t)) + \langle \nabla f(w'(t)), \bar{w}(t+1) - w'(t) \rangle + \frac{L_2}{2} \|\bar{w}(t+1) - w'(t)\|^2 \\
&\leq f(w'(t)) - \eta \|\nabla f(w'(t))\|^2 + \frac{L_2 B^2 (1+\gamma)^2}{2\mu^2} \|\nabla f(w'(t))\|^2 \\
&\leq f(w'(t)) - \left(\eta - \frac{L_2 B(1+\gamma)^2}{2\mu^2}\right) \|\nabla f(w'(t))\|^2,
\end{aligned} \tag{27}$$

where $\gamma = \max_{k,t}\{\gamma_k^{t+1}\}$.

In practice, we randomly select $K$ clients (i.e. $S_t$) to participate in each round of FL training, rather than involving all clients. Therefore, we consider the following error analysis:

$$e_t = \mathbb{E}_{S_t}\left[f\left(w\left(t+1\right)\right) - f\left(\bar{w}\left(t+1\right)\right)\right], \tag{28}$$

where $w(t+1)$ is the actual version of $\bar{w}(t+1)$, taking into account the randomness of client selection.

Based on Assumption B.5, it is easy to see that $f$ is also $L_1$-continuous. Therefore, we have

$$f\left(w(t+1)\right) \le f\left(\bar{w}(t+1)\right) + L_1 \left\|w(t+1) - \bar{w}(t+1)\right\| \tag{29}$$

Similarly, based on the $L_2$- smoothness of $f$, we have

$$
\begin{aligned}
L_1 &\le \left\|\nabla f\left(w'(t)\right)\right\| + L_2 \max\left(\left\|\bar{w}(t+1) - w'(t)\right\|, \left\|w(t+1) - w'(t)\right\|\right) \\
&\le \left\|\nabla f\left(w'(t)\right)\right\| + L_2 \left(\left\|\bar{w}(t+1) - w'(t)\right\| + \left\|w(t+1) - w'(t)\right\|\right)
\end{aligned}
\tag{30}
$$

Then, based on Eqs. (28), (29) and (30), we have

$$
\begin{aligned}
e_t &\le \mathbb{E}_{S_t}\left[L_1 \left\|w\left(t+1\right)\right) - \left(\bar{w}\left(t+1\right)\right\|\right] \\
&\le \mathbb{E}_{S_t}\left[\left(\left\|\nabla f\left(w'(t)\right)\right\| + L_2 \left(\left\|\bar{w}(t+1) - w'(t)\right\| + \left\|w(t+1) - w'(t)\right\|\right)\right) \cdot \left\|w(t+1) - \bar{w}(t+1)\right\|\right] \\
&= \left(\left\|\nabla f\left(w'(t)\right)\right\| + L_2 \left\|\bar{w}(t+1) - w'(t)\right\|\right) \cdot \mathbb{E}_{S_t}\left[\left\|w(t+1) - \bar{w}(t+1)\right\|\right] \\
&\quad + L_2\mathbb{E}_{S_t}\left[\left\|w(t+1) - w'(t)\right\| \cdot \left\|w(t+1) - \bar{w}(t+1)\right\|\right].
\end{aligned}
\tag{31}
$$

Moreover, we know that

$$
\begin{aligned}
&\mathbb{E}_{S_t}\left[\left\|w(t+1) - w'(t)\right\| \cdot \left\|w(t+1) - \bar{w}(t+1)\right\|\right] \\
&\le \mathbb{E}_{S_t}\left[\left(\left\|w(t+1) - \bar{w}\left(t+1\right)\right\| + \left\|\bar{w}(t+1) - w'(t)\right\|\right) \cdot \left\|w(t+1) - \bar{w}(t+1)\right\|\right] \\
&= \left\|\bar{w}(t+1) - w'(t)\right\| \cdot \mathbb{E}_{S_t}\left[\left\|w(t+1) - \bar{w}(t+1)\right\|\right] + \mathbb{E}_{S_t}\left[\left\|w(t+1) - \bar{w}(t+1)\right\|^2\right]
\end{aligned}
\tag{32}
$$

Overall, based on Eq. (31) and Eq. (32), we have

$$
\begin{aligned}
e_t &\le \left(\left\|\nabla f\left(w'(t)\right)\right\| + 2L_2 \left\|\bar{w}(t+1) - w'(t)\right\|\right) \cdot \mathbb{E}_{S_t}\left[\left\|w(t+1) - \bar{w}(t+1)\right\|\right] \\
&\quad + L_2\mathbb{E}_{S_t}\left[\left\|w(t+1) - \bar{w}(t+1)\right\|^2\right]
\end{aligned}
\tag{33}
$$

Similar to (Li et al., 2020), we provide the bounded variance of $w_k(t+1)$ as follows,

$$
\begin{aligned}
\mathbb{E}_{S_t}\left[\left\|w(t+1) - \bar{w}(t+1)\right\|^2\right] &\le \frac{1}{K}\mathbb{E}_k\left[\left\|w_k(t+1) - \bar{w}(t+1)\right\|^2\right] \\
&\le \frac{2}{K}\mathbb{E}_k\left[\left\|w_k(t+1) - w'(t)\right\|^2\right] \\
&\le \frac{2}{K}\frac{(1+\gamma)^2}{\mu^2}\mathbb{E}_k\left[\left\|\nabla f_k\left(w'(t)\right)\right\|^2\right] \\
&\le \frac{2}{K}\frac{(1+\gamma)^2 B^2}{\mu^2}\left\|\nabla f\left(w'(t)\right)\right\|^2.
\end{aligned}
\tag{34}
$$

Based on Jensen's inequality and Eq. (34), we have

$$
\begin{aligned}
\mathbb{E}_{S_t}\left[\left\|w(t+1) - \bar{w}(t+1)\right\|\right] &\le \sqrt{\mathbb{E}_{S_t}\left[\left\|w(t+1) - \bar{w}(t+1)\right\|^2\right]} \\
&\le \sqrt{\frac{2}{K}}\frac{(1+\gamma)B}{\mu}\left\|\nabla f\left(w'(t)\right)\right\|.
\end{aligned}
\tag{35}
$$

Combining Eq. (33) with Eqs. (25), (34), and (35), we have

$$
\begin{aligned}
e_t &\leq \left[ \left\| \nabla f\left(w'(t)\right) \right\| + 2L_2 \frac{(1+\gamma)B}{\mu} \left\| \nabla f\left(w'(t)\right) \right\| \right] \cdot \sqrt{\frac{2}{K}} \frac{(1+\gamma)B}{\mu} \left\| \nabla f\left(w'(t)\right) \right\| + \frac{2L_2}{K} \frac{(1+\gamma)^2 B^2}{\mu^2} \left\| \nabla f\left(w'(t)\right) \right\|^2 \\
&\leq \left[ \frac{\sqrt{2}\left(\mu B(1+\gamma) + 2L_2(1+\gamma)^2 B^2\right)}{\sqrt{K}\mu^2} + \frac{2L_2(1+\gamma)^2 B^2}{K\mu^2} \right] \left\| \nabla f\left(w'(t)\right) \right\|^2 .
\end{aligned}
$$
(36)

Based on Eq. (27) and Eq. (36), we have

$$
\begin{aligned}
\mathbb{E}_{S_t}\left[f\left(w(t+1)\right)\right] &\leq f\left(w'(t)\right) \\
&- \left\{ \eta - \frac{L_2 B^2 (1+\gamma)^2}{2\mu^2} - \left[ \frac{\sqrt{2}\left(\mu B(1+\gamma) + 2L_2(1+\gamma)^2 B^2\right)}{\sqrt{K}\mu^2} + \frac{2L_2(1+\gamma)^2 B^2}{K\mu^2} \right] \right\} \left\| \nabla f\left(w'(t)\right) \right\|^2 .
\end{aligned}
$$
(37)

Then, we set $m = \eta - \frac{L_2 B^2 (1+\gamma)^2}{2\mu^2} - \left[ \frac{\sqrt{2}\left(\mu B(1+\gamma) + 2L_2(1+\gamma)^2 B^2\right)}{\sqrt{K}\mu^2} + \frac{2L_2(1+\gamma)^2 B^2}{K\mu^2} \right]$ and we have

$$
\mathbb{E}_{S_t}\left[f\left(w(t+1)\right)\right] \leq f\left(w'(t)\right) - m \left\| \nabla f\left(w'(t)\right) \right\|^2 .
$$
(38)

Considering **DP mechanism** and **our low-rank proximal optimization**:

Based on the $L_1$-continuity of $f$, we have

$$
f\left(w'(t+1)\right) \leq f\left(w(t+1)\right) + L_1 \left\| w'(t+1) - w(t+1) \right\|
$$
(39)

Thus,

$$
\begin{aligned}
\mathbb{E}_{S_t}\left[f\left(w'(t+1)\right)\right] - \mathbb{E}_{S_t}\left[f\left(w(t+1)\right)\right] &\leq L_1 \mathbb{E}_{S_t}\left[\left\| w'(t+1) - w(t+1) \right\|\right] \\
&= \frac{L_1}{K} \sum_{k \in S_t} \left[\left\| \hat{w}_k(t+1) - w_k(t+1) \right\|\right] \\
&\leq \frac{\sqrt{2}L_1}{K} \left\| \hat{\mathcal{W}} - \mathcal{W} \right\|_F ,
\end{aligned}
$$
(40)

where $\hat{\mathcal{W}} = \mathrm{fold}\left([\hat{w}_1(t), \cdots, \hat{w}_K(t)]^T\right)$ and $\mathcal{W} = \mathrm{fold}\left([w_1(t), \cdots, w_K(t)]^T\right) \in \mathbb{R}^{d \times h \times K}$.

Next, we know that

$$
\left\| \hat{\mathcal{W}} - \mathcal{W} \right\|_F \leq \left[ \left\| \hat{\mathcal{W}} - \mathcal{W}_{\mathcal{N}} \right\|_F + \left\| \mathcal{W}_{\mathcal{N}} - \mathcal{W} \right\|_F \right],
$$
(41)

where $\mathcal{W}_{\mathcal{N}} = \mathrm{fold}\left([w'_1(t), \cdots, w'_K(t)]^T\right) \in \mathbb{R}^{d \times h \times K}$.

Based on Parseval's theorem and Algorithm A.4, we have

$$
\begin{aligned}
\left\| \hat{\mathcal{W}} - \mathcal{W}_{\mathcal{N}} \right\|_F &= \left\| \overline{\hat{\mathcal{W}}} - \overline{\mathcal{W}}_N \right\|_F \\
&\leq \sum_{i=1}^{K} \left\| \overline{\hat{\mathbf{W}}}^{(i)} - \overline{\mathbf{W}}_N^{(i)} \right\|_F \\
&= \sum_{i=1}^{K} \left\| \overline{\mathbf{U}}^{(i)} \cdot \mathrm{diag}\left(\{\max(\sigma_j - \tau, 0)\}_{j=1}^r\right)^{(i)} \cdot \overline{\mathbf{V}}^{(i)} - \overline{\mathbf{U}}^{(i)} \cdot \mathrm{diag}\left(\{\sigma_j\}_{j=1}^r\right)^{(i)} \cdot \overline{\mathbf{V}}^{(i)} \right\|_F \\
&= \sum_{i=1}^{K} \left\| \mathrm{diag}\left(\{\max(\sigma_j - \tau, 0)\}_{j=1}^r\right)^{(i)} - \mathrm{diag}\left(\{\sigma_j\}_{j=1}^r\right)^{(i)} \right\|_F \\
&\leq \sqrt{r}\tau_0, (\text{as} \quad \tau = \tau_0/K)
\end{aligned}
$$
(42)

where $\{\sigma_j\}_{j=1}^r$ are the singular values of $\mathcal{W}_{\mathcal{N}}$ and $\tau$ is the truncated threshold of Algorithm A.4.

Based on our DP with the Gaussian mechanism, we have

$$\|\mathcal{W}_{\mathcal{N}} - \mathcal{W}\|_F = \mathbb{E}\left[\eta \cdot \|\mathcal{N}\|_F\right] = \sqrt{d}\sqrt{h}C\sigma, \tag{43}$$

where $\mathcal{N} \sim \mathcal{N}(0, \mathcal{I}\sigma^2 C^2/K)$ and $\mathcal{I} \in \mathbb{R}^{d \times h \times K}$.

**Choice of $\tau$**    When the number of clients $K$ is large, and the variance of local Gaussian noise $\mathcal{N}$ is small, we set a smaller truncation threshold with the choice $\tau = \tau_0/K$. This allows for the retention of more accurate semantic information from individual clients.

Combining Eq. (40) with Eqs. (41), (42), (42), we have

$$\mathbb{E}_{S_t}\left[f\left(w'(t+1)\right)\right] - \mathbb{E}_{S_t}\left[f\left(w(t+1)\right)\right] \leq \frac{\sqrt{2}L_1}{K}\left(\sqrt{r}\tau_0 + \sqrt{d}\sqrt{h}C\sigma\right). \tag{44}$$

Overall, based on Eq. (38) and Eq. (44), we have

$$f\left(w'(t+1)\right) \leq f\left(w'(t)\right) - m\left\|\nabla f\left(w'(t)\right)\right\|^2 + \frac{\sqrt{2}L_1}{K}\left(\sqrt{r}\tau_0 + \sqrt{d}\sqrt{h}C\sigma\right). \tag{45}$$

Based on the $\mu$-strongly convexity of $f$, we have

$$f(w(t)) \geq f(w'(t)) + \nabla f(w'(t))^T(w(t) - w'(t)) + \frac{\mu}{2}\|w(t) - w'(t)\|^2. \tag{46}$$

Now, minimize the inequity with respect to $w(t)$ and we have

$$f\left(w^*\right) \geq f(w'(t)) - \frac{1}{2\mu}\|\nabla f(w'(t))\|^2, \tag{47}$$

where $w^*$ is the convergent global model of $w(t)$.

Based on Eq. (45) and Eq. (47), we have

$$f\left(w'(t+1)\right) \leq f\left(w'(t)\right) - 2\mu m\left(f\left(w'(t)\right) - f\left(w^*\right)\right) + \frac{\sqrt{2}L_1}{K}\left(\sqrt{r}\tau_0 + \sqrt{d}\sqrt{h}C\sigma\right), \tag{48}$$

and thus

$$f\left(w'(t+1)\right) - f\left(w^*\right) \leq (1 - 2\mu m)\left(f\left(w'(t)\right) - f\left(w^*\right)\right) + \frac{\sqrt{2}L_1}{K}\left(\sqrt{r}\tau_0 + \sqrt{d}\sqrt{h}C\sigma\right). \tag{49}$$

Taking $t$ from 0 to $T-1$ in Eq. (49),

$$f\left(w'(1)\right) - f\left(w^*\right) \leq (1 - 2\mu m)\left(f\left(w'(0)\right) - f\left(w^*\right)\right) + \frac{\sqrt{2}L_1}{K}\left(\sqrt{r}\tau_0 + \sqrt{d}\sqrt{h}C\sigma\right). \tag{50}$$

$$f\left(w'(2)\right) - f\left(w^*\right) \leq (1 - 2\mu m)\left(f\left(w'(1)\right) - f\left(w^*\right)\right) + \frac{\sqrt{2}L_1}{K}\left(\sqrt{r}\tau_0 + \sqrt{d}\sqrt{h}C\sigma\right). \tag{51}$$

$$\cdots$$

$$f\left(w'(T)\right) - f\left(w^*\right) \leq (1 - 2\mu m)\left(f\left(w'(T-1)\right) - f\left(w^*\right)\right) + \frac{\sqrt{2}L_1}{K}\left(\sqrt{r}\tau_0 + \sqrt{d}\sqrt{h}C\sigma\right). \tag{52}$$

and subsequently substituting each resulting expression one by one, we obtain

$$f\left(w'(T)\right) - f\left(w^*\right) \leq (1 - 2\mu m)^T\left(f\left(w'(1)\right) - f\left(w^*\right)\right) + \frac{\sqrt{2}L_1}{K}\left(\sqrt{r}\tau_0 + \sqrt{d}\sqrt{h}C\sigma\right)\frac{1 - (1 - 2\mu m)^T}{2\mu m}. \tag{53}$$

When selecting a sufficiently large $T$ and satisfying $0 < m < 1/\mu$, we have

$$\epsilon_u = f\left(w'\right) - f\left(w^*\right) \leq \frac{\sqrt{2}L_1}{K}\left(\sqrt{r}\tau_0 + \sqrt{d}\sqrt{h}C\sigma\right)\frac{1}{2\mu m}. \tag{54}$$

where $w'$ is the convergent global model of $w'(t)$.

Overall, with the choice of $m, \tau$ and $T$ specified in the theorem, we have

$$\epsilon_u = O(\frac{\sqrt{r} + \sqrt{d}}{K}), \tag{55}$$

where $r$ is the rank of the parameter tensor after our processing and $d$ is the dimension of input data.

Especially, When we choose an appropriate truncation threshold (regularization coefficient) such that the resulting parameter tensor is **low-rank**, we have

$$\epsilon_u = O(\frac{\sqrt{d}}{K}). \tag{56}$$

$\square$

### B.4. Proof of Theorem 4.5 (Privacy Analysis of Our FedCEO)

*Proof.* Let $\mathcal{M} : \mathcal{D} \to \mathcal{R}$ denote algorithm 1 that satisfies user-level $(\epsilon, \delta)$-DP. Based on its definition, we know for any two adjacent datasets $D, D' \in \mathcal{D}$ that differ by an individual user's data and all outputs $S \subseteq \mathcal{R}$ it holds that

$$\Pr[\mathcal{M}(D) \in S] \le e^\epsilon \Pr[\mathcal{M}(D') \in S] + \delta. \tag{57}$$

By theorem 3.1 and Algorithm A.4, we know our low-rank proximal optimization is equivalent to the truncated tSVD algorithm, so it is a deterministic function, denoted as $\mathcal{F} : \mathcal{R} \to \mathcal{R}'$.

Fix any pair of neighboring datasets $\mathcal{D}, \mathcal{D}'$ with $\|\mathcal{D} - \mathcal{D}'\| \le 1$, and fix any output $S \subseteq \mathcal{R}'$. Let $Z = \{z \in \mathcal{R} | \mathcal{F}(z) \in S\}$, we have

$$
\begin{aligned}
\Pr[\mathcal{F}(\mathcal{M}(\mathcal{D})) \in S] &= \Pr[\mathcal{M}(\mathcal{D}) \in Z] \\
&\le \exp(\epsilon) \Pr[\mathcal{M}(\mathcal{D}') \in Z] + \delta \\
&= \exp(\epsilon) \Pr[\mathcal{F}(\mathcal{M}(\mathcal{D}')) \in S] + \delta
\end{aligned}
\tag{58}
$$

$\square$

## C. Experiments Setup and More Results

### C.1. Local Model and Hyperparameters

**Models**. In the paper, we employ a two-layer MLP for the EMNIST dataset and a LeNet-5 for the CIFAR-10 dataset. The specific network architectures are as follows.

**MLP**:

(1) (input layer): Linear(in_features=$d$, out_features=64, bias=False)

(2) (dropout layer): Dropout($p$=0.5, inplace=False)

(3) (activation layer): ReLU()

(4) (hidden layer): Linear(in_features=64, out_features=num_classes, bias=False)

(5) (activation layer): Softmax(dim=1)

**LeNet-5**:

(1) (conv1): Conv2d(3, 32, kernel_size=(5, 5), stride=(1, 1))

(2) (pool): MaxPool2d(kernel_size=2, stride=2, padding=0, dilation=1, ceil_mode=False)

(3) (conv2): Conv2d(32, 64, kernel_size=(5, 5), stride=(1, 1))

(4) (pool): MaxPool2d(kernel_size=2, stride=2, padding=0, dilation=1, ceil_mode=False)

(5) (fc1): Linear(in_features=1600, out_features=512, bias=True)

(6) (activation layer): ReLU()

(7) (fc2): Linear(in_features=512, out_features=512, bias=True)

(8) (activation layer): ReLU()

(9) (fc3): Linear(in_features=512, out_features=10, bias=True)

**Hyperparameters**. For each dataset, we uniformly set the global communication rounds $T$ to 300, the total number of clients $N$ to 100, and the sampled client number $K$ to 10, resulting in a sampling rate $p$ of 0.1. Local training employs stochastic gradient descent (SGD) with 30 epochs ($E$), a learning rate $\eta$ of 0.1, and a batch size $B$ of 64.

Other personalized hyperparameters: initial coefficient $\lambda$, common ratio $\vartheta$, and interval $I$ (searched within a grid range), whose values can be qualitatively guided by practical application scenarios, and their impact on model utility is robust. For $\lambda$, the search range is [55, 100, 5] when $\sigma_g = 1.0$; [0.1, 10, log] when $\sigma_g = 1.5$; and [0.01, 1, log] when $\sigma_g = 2.0$. For scenarios with stricter privacy requirements (larger noise), we need to choose a smaller $\lambda$ to achieve a smoother semantic space. For $\vartheta$, the search range is [1.01, 1.10, 0.01]. For $I$, the search range is [10, 15, 20, 25, 30]. Their specific values are listed in Table 2.

Table 2. Our focused hyperparameters for three privacy settings on EMNIST and CIFAR-10.

| Dataset | Model | Setting | Hyperparameter |
|---|---|---|---|
| EMNIST | MLP-2-Layers | $\sigma_g = 1.0$ | $\lambda = 70, \vartheta = 1.08, I = 30$ |
| | | $\sigma_g = 1.5$ | $\lambda = 0.5, \vartheta = 1.04, I = 20$ |
| | | $\sigma_g = 2.0$ | $\lambda = 0.03, \vartheta = 1.06, I = 20$ |
| CIFAR-10 | LeNet-5 | $\sigma_g = 1.0$ | $\lambda = 85, \vartheta = 1.03, I = 15$ |
| | | $\sigma_g = 1.5$ | $\lambda = 10, \vartheta = 1.07, I = 10$ |
| | | $\sigma_g = 2.0$ | $\lambda = 0.6, \vartheta = 1.04, I = 10$ |

Partial parameter analysis results are as follows:

Table 3. Testing accuracy on CIFAR-10 of different intervals $I$ under various privacy settings with three common $\sigma_g$.

| Dataset | Model | Setting | $I = 10$ | $I = 15$ | $I = 20$ | $I = 25$ | $I = 30$ |
|---|---|---|---|---|---|---|---|
| CIFAR-10 | LetNet-5 | $\sigma_g = 1.0$ | 53.62% | **54.16%** | 53.01% | 52.81% | 52.80% |
| | | $\sigma_g = 1.5$ | **50.00%** | 49.71% | 48.90% | 47.41% | 47.98% |
| | | $\sigma_g = 2.0$ | **45.35%** | 44.37% | 44.81% | 43.75% | 43.20% |

## C.2. More Empirical Results

### C.2.1. MODEL TRAINING EFFICIENCY

To validate the efficiency of our server-side low-rank proximal optimization, we conduct a comparative analysis of the runtime between our FedCEO and other methods in Table 4. We observe that the training efficiency of our method significantly surpasses PPSGD and CENTAUR, approaching the efficiency of UDP-FedAvg without any utility improvement.

Table 4. Runtime for our FedCEO and other methods on EMNIST and CIFAR-10 (One NVIDIA GeForce RTX 4090).

| Time / h | UDP-FedAvg | PPSGD | CENTAUR | FedCEO |
|---|---|---|---|---|
| EMNIST | 5.436 | > 24 | > 24 | 5.445 |
| CIFAR-10 | 3.876 | > 24 | > 24 | 3.908 |

### C.2.2. UTILITY FOR HETEROGENEOUS FL SETTINGS

To validate the effectiveness of our model in heterogeneous federated learning (Li et al., 2020; Huang et al., 2025; Fu et al., 2025), we conduct experiments using an MLP as the local model on CIFAR-10 in Table 5. We report the testing accuracy for

both iid and non-iid [1] scenarios. It can be observed that our FedCEO maintains state-of-the-art performance. Additionally, in non-iid scenarios, we typically choose a larger $\lambda$ to reduce the global semantic smoothness, preserving more personalized local information.

*Table 5.* Testing accuracy (%) on CIFAR-10 under iid setting and non-iid setting.

| Dataset | Heterogeneity | Setting | UDP-FedAvg | PPSGD | CENTAUR | FedCEO |
|---------|---------------|---------|------------|-------|---------|--------|
| CIFAR-10 | iid | $\sigma_g = 1.0$ | 40.21% | 41.34% | 42.17% | **42.76%** |
| | | $\sigma_g = 1.5$ | 35.79% | 37.28% | 38.21% | **39.16%** |
| | | $\sigma_g = 2.0$ | 31.62% | 33.51% | 33.86% | **35.93%** |
| | non-iid | $\sigma_g = 1.0$ | 33.09% | 34.91% | 34.56% | **36.10%** |
| | | $\sigma_g = 1.5$ | 28.92% | 31.40% | 30.87% | **32.39%** |
| | | $\sigma_g = 2.0$ | 26.54% | 28.01% | 28.11% | **29.13%** |

### C.2.3. UTILITY FOR OTHER LOCAL ARCHITECTURE AND DATASET

To further validate the applicability of our framework, we conduct experiments using more complex local architectures and other types of datasets, as shown in Table 6. Specifically, we use AlexNet as the local model on CIFAR-10 and LSTM on the text dataset Sentiment140 (Sent140) (Caldas et al., 2018). It can be observed that our FedCEO still maintains SOTA performance.

*Table 6.* Testing accuracy (%) on CIFAR-10 and Sent140 under $\delta = 10^{-5}$ and various privacy settings with three common $\sigma_g$.

| Dataset | Model | Setting | UDP-FedAvg | PPSGD | CENTAUR | FedCEO |
|---------|-------|---------|------------|-------|---------|--------|
| CIFAR-10 | AlexNet | $\sigma_g = 1.0$ | 50.67% | 56.58% | 58.44% | **60.73%** |
| | | $\sigma_g = 1.5$ | 41.11% | 51.07% | 50.20% | **55.49%** |
| | | $\sigma_g = 2.0$ | 33.38% | 39.93% | 43.95% | **49.06%** |
| Sent140 | LSTM | $\sigma_g = 1.0$ | 60.31% | 61.04% | 63.33% | **65.70%** |
| | | $\sigma_g = 1.5$ | 57.62% | 57.87% | 59.05% | **60.22%** |
| | | $\sigma_g = 2.0$ | 50.94% | 55.12% | 54.88% | **56.65%** |

### C.2.4. MORE DETAILS FOR PRIVACY EXPERIMENTS

In the three federated learning frameworks, we consider a semi-honest adversary at the server, engaging in a gradient inversion attack on the model (gradient) uploaded by a specific client in a given round. This adversarial action is based on the DLG algorithm, and the detailed attack procedure is presented in Figure 6 to 8.

---

[1] **non-iid** means the data among clients are not independent and identically distributed (**iid**).

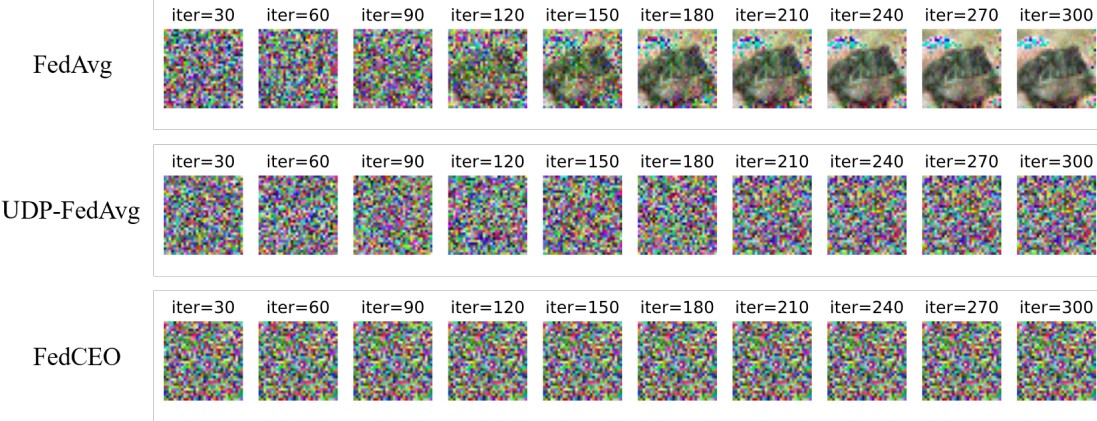

*Figure 6.* Privacy attack process on three FL frameworks based on DLG. We set the attack target as the image with the index 25 in CIFAR-10. For UDP-FedAvg and our FedCEO, we configure the local model as LeNet with a noise parameter of $\sigma_g = 1.0$.

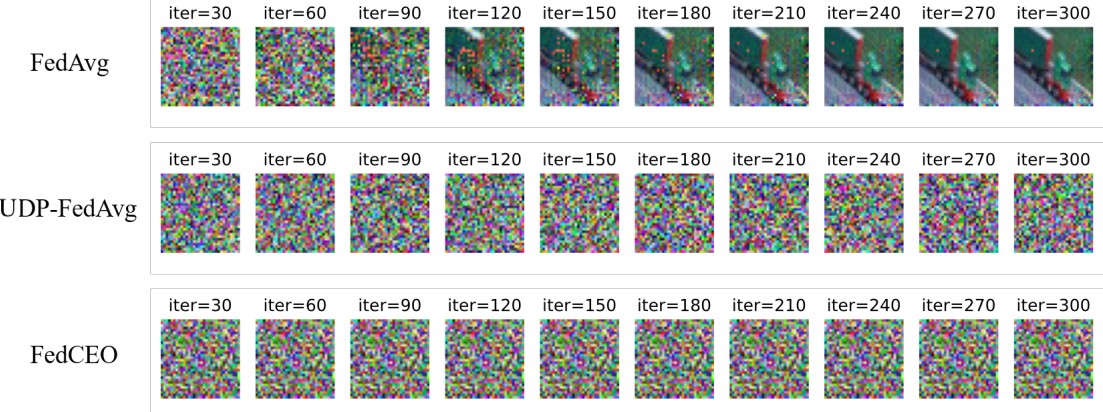

*Figure 7.* Privacy attack process on three FL frameworks based on DLG. We set the attack target as the image with the index 50 in CIFAR-10. For UDP-FedAvg and our FedCEO, we configure the local model as LeNet with a noise parameter of $\sigma_g = 1.5$.

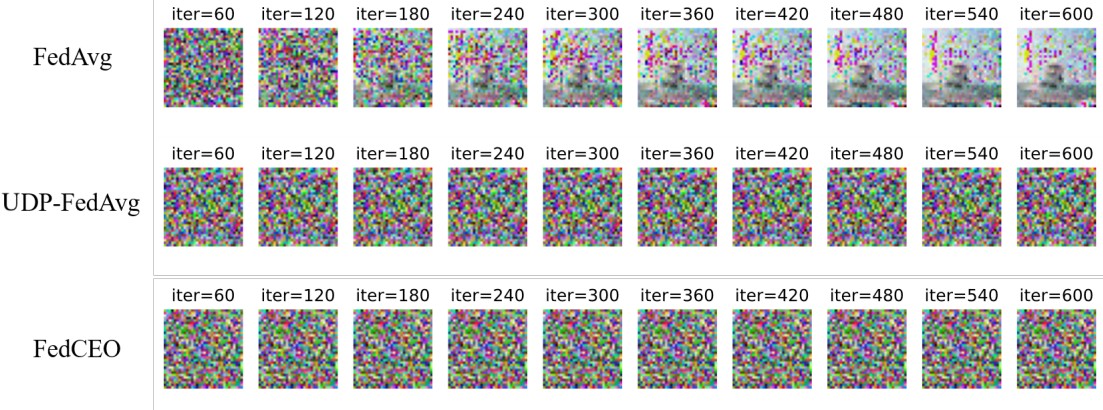

*Figure 8.* Privacy attack process on three FL frameworks based on DLG. We set the attack target as the image with the index 100 in CIFAR-10. For UDP-FedAvg and our FedCEO, we configure the local model as LeNet with a noise parameter of $\sigma_g = 2.0$.

