# OpenReview forum: "Clients Collaborate: Flexible Differentially Private Federated Learning with Guaranteed Improvement of Utility-Privacy Trade-off"
_ICML.cc/2025/Conference — ICML 2025 poster_

### Official Review · Reviewer_vBZu · 2025-03-12

**Overall Recommendation:** 3

**Summary:**

The paper proposes a novel federated learning framework, FedCEO, aimed at balancing model utility and user privacy through collaboration among clients. The authors introduce a compelling case study to illustrate the potential of semantic collaboration among clients in enhancing the utility of the global model, and they construct the update process as a high-order tensor low-rank optimization. Furthermore, the authors theoretically prove that their model achieves a $\sqrt{d}$-order improvement in the utility-privacy trade-off bound. Extensive experiments, covering both utility and privacy aspects, demonstrate the effectiveness of FedCEO.

**Claims And Evidence:**

The claims made in the submission are well-supported by clear and convincing evidence. The authors provide thorough theoretical analysis and experimental validation to substantiate the significant improvements in the utility-privacy trade-off achieved by the FedCEO. Firstly, the authors derive the utility-privacy trade-off bound for FedCEO in the theoretical section and prove its improvement by a factor of $\sqrt{d}$ over existing techniques. Secondly, extensive experiments on multiple representative datasets validate the performance enhancements and privacy-preserving capabilities of FedCEO under various privacy settings.

**Essential References Not Discussed:**

The references in the paper already cover most of the essential related works

**Experimental Designs Or Analyses:**

The experimental designs and analyses in the paper have been carefully reviewed and are generally sound and valid. The experimental section covers a comprehensive evaluation of model utility, privacy protection, and the utility-privacy trade-off, with a well-designed and convincing approach. Particularly, the authors further validate the robustness of FedCEO in privacy protection through gradient inversion attack experiments, with clear and reproducible results. Overall, the experimental designs and analyses are rigorous and support the main conclusions of the paper.

**Methods And Evaluation Criteria:**

The proposed methods and evaluation criteria are highly appropriate and well-suited for the problem at hand. The experiments utilize representative datasets such as CIFAR-10, EMNIST and Sent140, along with various model architectures (e.g., MLP, LeNet and AlexNet), thoroughly validating the method's effectiveness and generality. Additionally, the authors further validate privacy protection through gradient inversion attack experiments, making the evaluation criteria comprehensive and convincing.

**Other Comments Or Suggestions:**

Please refer to the weaknesses.

**Other Strengths And Weaknesses:**

Strengths:
The strengths of the paper lie in its originality and practical applicability. Firstly, the proposed framework addresses the utility-privacy trade-off in differentially private federated learning through semantic collaboration among clients, which is highly innovative. Secondly, the paper not only theoretically proves the improvement in the utility-privacy trade-off bound but also validates its effectiveness on real-world datasets through extensive experiments, demonstrating its potential in practical scenarios.

Weaknesses:
The weaknesses of the paper can be outlined as follows:
1. Lack of Depth in Technical Details: The detailed discussions on parameter selection are somewhat brief in the main text. More technical details and discussions on parameter tuning could further enhance the reproducibility and practicality of the paper.
2. Unclear Descriptions of Concepts: Some concepts or keywords in the paper, such as "semantic collaboration" and "global semantic space smoothing," could be described more clearly and in greater detail. More precise definitions and explanations would help readers better understand these concepts.
3. Breadth of Comparative Experiments: Although the paper compares with some existing methods, the scope of comparative experiments could be expanded to include more recent federated learning and differential privacy methods. This would provide a more comprehensive demonstration of FedCEO's advantages.

**Questions For Authors:**

1. Could the authors further explain why FedCEO is referred to as a "flexible" DPFL framework, as indicated in the title?
2. What is the definition of "global semantic space" in the paper?
3. How exactly is the non-iid data distribution set up in Table 5?

**Relation To Broader Scientific Literature:**

The key contributions of the paper are closely related to the broader scientific literature, particularly in the fields of DPFL. Firstly, the FedCEO addresses the utility-privacy trade-off in differentially private federated learning through semantic collaboration among clients, which differs from existing approaches that focus on regularization or personalization to improve model utility. For example, *Cheng et al.* proposed local update regularization and sparsification techniques, while *PPSGD* improved model utility through a personalized privacy-preserving stochastic gradient optimization algorithm. However, these methods primarily focus on constraining local updates and do not fully leverage the semantic complementarity among clients. FedCEO further enhances model utility through high-order tensor low-rank optimization and theoretically proves its improvement in the utility-privacy trade-off bound. Additionally, the experimental results of this paper demonstrate significant performance improvements and strict privacy guarantees compared to existing literature (e.g., *CENTAUR*). Overall, the paper introduces an innovative approach based on existing research and provides a new solution to the utility-privacy trade-off problem in federated learning.

**Theoretical Claims:**

The theoretical proofs in the paper have been carefully reviewed and are generally correct and rigorous.

---

> ### Author Rebuttal · Authors · 2025-04-01
>
> We sincerely appreciate the reviewer’s positive feedback on our theoretical and empirical contributions. Below are our point-to-point responses to the raised concerns and suggestions.
>
> > **W1**: Lack of Depth in Technical Details: The detailed discussions on parameter selection are somewhat brief in the main text. More technical details and discussions on parameter tuning could further enhance the reproducibility and practicality of the paper.
>
> The reviewer noted brevity in parameter selection discussions. We have detailed hyperparameter search ranges and final configurations in **Appendix C.1 (Table 2)**. For example, on CIFAR-10 with σ_g=1.5, we set λ=10, ϑ=1.07, and I=10. These parameters are selected via grid search and adaptively adjusted based on privacy requirements (e.g., smaller λ for stricter privacy). Please refer to the response to the **reviewer RVpv (Experimental Designs/Analyses) for additional experiments**.
>
> > **W2 & Q2**: Unclear Descriptions of Concepts: Some concepts or keywords in the paper, such as "semantic collaboration" and "global semantic space smoothing," could be described more clearly and in greater detail. More precise definitions and explanations would help readers better understand these concepts.
>
> - **Global Semantic Space Smoothing**: Refers to the low-rank representation of client parameters in the spectral space after tensor decomposition, where high-frequency components are truncated to preserve semantic correlations
>
> - **Semantic Collaboration**: Achieved by integrating complementary semantic information across clients via low-rank optimization (Section 3.2). Please refer to the responses to **Reviewer kpz4 (Q2) for specific examples**.
>
> > **W3**: Breadth of Comparative Experiments: Although the paper compares with some existing methods, the scope of comparative experiments could be expanded to include more recent federated learning and differential privacy methods. This would provide a more comprehensive demonstration of FedCEO's advantages.
>
> We appreciate your suggestion and will incorporate additional baseline comparisons in the revised manuscript to provide a more comprehensive validation of our findings.
>
> > **Q1**: Could the authors further explain why FedCEO is referred to as a "flexible" DPFL framework, as indicated in the title?
>
> As illustrated in Figure 2, our FedCEO can flexibly adapt to different privacy settings and noise accumulation during continuous training by setting different initial values and employing an adaptive thresholding rule based on a geometric series. The specific advantages of this "flexibility" are demonstrated in the experimental results in Table 1.
>
> > **Q3**: How exactly is the non-iid data distribution set up in Table 5?
>
> Apologies for the lack of details on the degree of non-iid data. For non-iid data, we used a common method to construct data heterogeneity.
>
> Taking CIFAR-10 as an example:
>
> - First, we sorted all images in the dataset by their labels.
>
> - Then, for each client, we randomly selected 250 consecutive images twice. Since CIFAR-10 has ten categories, each client has at least one category and at most four categories, verifying that our framework works even in highly heterogeneous situations.

---

### Official Review · Reviewer_kpz4 · 2025-03-12

**Overall Recommendation:** 4

**Summary:**

The authors propose a novel federated learning framework with the differential privacy mechanism, focusing on improving the trade-off between model utility and user privacy. By leveraging tensor decomposition techniques, the proposed method can model the dynamic semantic relationships among different clients, thereby enhancing the performance of the global model. The authors provide comprehensive mathematical analysis and empirical evidence, demonstrating that their framework achieves a better privacy-utility trade-off.

**Claims And Evidence:**

Yes, the main claims in the paper are well supported by both experimental results and theoretical analysis. The visualization experiment in Figure 1 and Theorem 3.1 support the motivation of this work. Corollary 4.6 and the experiments in Figure 4 provide strong evidence demonstrating the effectiveness of the proposed framework in improving the privacy-utility trade-off in federated learning.

**Essential References Not Discussed:**

The references in this work are generally comprehensive. However, it would be beneficial to include a discussion on recent related works, such as Provable Mutual Benefits from Federated Learning in Privacy-Sensitive Domains (AISTATS 2024).

**Experimental Designs Or Analyses:**

The experiments in this work are rigorous and comprehensive, systematically designed to evaluate utility and privacy. Additionally, the study includes experiments under heterogeneous federated settings, efficiency comparisons, and tests on textual datasets.

**Methods And Evaluation Criteria:**

The proposed method is appropriate, as Proposition 3.2 in the paper proves that the classic federated learning algorithm FedAvg is a special case of it. The chosen datasets are suitable and comprehensive, covering general datasets of different modalities and adopting common federated learning data partitioning methods.

**Other Comments Or Suggestions:**

1. It is recommended to compress the length of the background knowledge in Section 2 and place more emphasis on the description of the method.
2. The description of the experiments in the abstract needs to be revised, as the paper considers both image and text datasets.
3. In Section 1.1 (Related Work), it is suggested to discuss the differences between the proposed method and the latest techniques in terms of mechanism design and theoretical analysis. Additionally, it might be useful to add comparative baselines.

**Other Strengths And Weaknesses:**

**Strengths:**
1. This paper primarily focuses on achieving the utility-privacy trade-off in differentially private federated learning framework. This is a crucial research topic with significant implications for the practical deployment of federated learning algorithms in industrial applications.
2. This work follows a rigorous logical structure, where the authors naturally introduce their framework through preliminary experiments and insights from previous works.
3. The proposed approach is novel, as it connects tensor decomposition algorithms with federated model updates and provides an equivalence proof. This offers new perspectives for designing global model update paradigms in federated learning.
4. Theoretical analysis is thorough, and the final conclusions demonstrate the effectiveness of low-rank modeling, providing readers with a deeper understanding of the fundamental principles behind the proposed approach.

**Weaknesses:**
1. There is still room for improvement in the paper’s writing, particularly in terms of the proportion of content across different sections.
2. The paper lacks a discussion of some recent related works, such as Provable Mutual Benefits from Federated Learning in Privacy-Sensitive Domains (AISTATS 2024).

**Questions For Authors:**

1. How should "semantic complementarity" in the introduction be understood?
2. Can you provide a specific example of how semantic collaboration between clients is implemented?
3. Apart from the T-tSVD algorithm used in the paper, can other more efficient tensor optimization methods be applied?
4. Can the model optimization in Equation 2 be understood as a denoising process of the noisy model tensor? Would this compromise the privacy of the overall differential privacy federated framework? I'm slightly concerned about this.

**Relation To Broader Scientific Literature:**

Previous works have considered spectral decomposition at the local client level, while this method further incorporates inter-client relationships and models them using high-order tensor singular value decomposition. Compared to (Jain et al., 2021) and CENTAUR (Shen et al., 2023), the theoretical bounds have been further optimized.

**Theoretical Claims:**

Yes, I mainly focus on the proof details of Theorem 4.3. The overall proof approach is sound and successfully demonstrates the effect of low-rank (semantic collaboration).

---

> ### Author Rebuttal · Authors · 2025-04-01
>
> We sincerely appreciate the reviewer’s constructive feedback and recognition of our work. Below are our point-by-point responses:
>
> > **W1 & Comments1, 2**
>
> Condensing Background Knowledge: We agree with the suggestion. In the revised manuscript, we will streamline Section 2 (Preliminaries) by moving detailed definitions to the appendix, allowing greater emphasis on the core methodology (e.g., the optimization objective of FedCEO in Section 3 and the motivation for low-rank modeling).
>
> Abstract Experiment Description: Thank you for the correction. We will update the abstract to include results on text datasets (e.g., Sentiment140): "...with experiments on representative image and text datasets…".
>
> > **W2 & Comments3**
>
> We will add a discussion of 'Provable Mutual Benefits from Federated Learning in Privacy-Sensitive Domains' in the Related Work (Section 1.1). This work proposes a theoretical framework for designing personalized privacy-preserving protocols that provably benefit all participants in privacy-sensitive domains. Compared to our approach, it does not explicitly provide quantitative utility-privacy bounds but focuses on existence proofs. Both works involve inter-client collaboration, representing two complementary approaches: **protocol design controlling client-side noise** versus **post-processing algorithms correcting inter-client semantics**.
>
> > **Q1**
>
> "Semantic complementarity" refers to the fact that local DP noise disrupts semantic information differently across clients. In DPFL, due to the randomness of noise introduced by each client, specific semantic information might be disrupted in some clients while remaining relatively intact in others, leading to semantic complementarity among different clients. By integrating the commonalities of parameters via tensor low-rank optimization, the server recovers global semantic smoothness. For example, in Figure 1 (CIFAR-10), the semantic space of the 10th class is corrupted (red row), but FedCEO leverages intact semantic features from other clients (e.g., client 7, 9) to restore smoothness (blue row), improving classification accuracy.
>
>
> > **Q2**
>
> Consider a DPFL framework with only two clients performing animal image classification. This collaboration mechanism enables semantic complementarity between the two clients' models, thereby improving the global model's performance in DPFL.
>
> For one client, the noise might severely corrupt parameters responsible for recognizing facial features, while for the other client, the noise might primarily distort parameters for limb recognition. However, by performing a Fourier transform, each slice will contain information from all clients. Subsequently, applying truncated SVD to all slices can be viewed as projecting the perturbed parameters back into a smooth semantic space, leveraging knowledge from all clients.
>
> Finally, after performing the inverse Fourier transform, each slice (representing a client's parameters) will have adaptively incorporated information from other clients while retaining some of its original knowledge.
>
> > **Q3**
>
> T-tSVD is chosen for its efficiency in dynamic thresholding via Fourier-domain matrix factorization (Table 4). Future work may explore other methods (e.g.,deep tensor learning), provided they align with privacy constraints. Existing attempts can be seen in our response to **Reviewer RVpv (W3 & Other Comments/Suggestions)**.
>
> > **Q4**
>
> Equation (2) is not a denoising process, as we do not utilize any prior information about the DP noise in our modeling. Essentially, it is a controlled fusion operation, and Proposition 3.2 demonstrates that FedAvg is in fact a special case of our FedCEO method—yet FedAvg is not considered to possess denoising capabilities. Furthermore, both Theorem 4.5 and the experiments in Section 5.3 confirm that the server-side operations in Algorithm 2 do not compromise the privacy guarantees of the DPFL framework.

---

> > ### Comment · Reviewer_kpz4 · 2025-04-03
> >
> > Thanks for the response. I still have some questions as follow:
> >
> > 1. You mentioned in your response that “due to the randomness of noise introduced by each client, specific semantic information might be disrupted in some clients while remaining relatively intact in others, leading to semantic complementarity among different clients”. Does this imply that the method assumes a certain degree of IID data, and its effectiveness would decline when client data distributions are highly heterogeneous?
> >
> > 2. In Q3, you mentioned exploring alternative tensor optimization methods (e.g., deep tensor learning). Could these methods potentially leak client privacy due to additional parameters or nonlinear operations? How would you ensure compatibility with existing privacy constraints?

---

> > > ### Author Response · Authors · 2025-04-06
> > >
> > > We sincerely appreciate your thoughtful questions and the opportunity to clarify these important aspects of our work.
> > >
> > > > You mentioned in your response that “due to the randomness of noise introduced by each client, specific semantic information might be disrupted in some clients while remaining relatively intact in others, leading to semantic complementarity among different clients”. Does this imply that the method assumes a certain degree of IID data, and its effectiveness would decline when client data distributions are highly heterogeneous?
> > >
> > > The FedCEO framework does **not** inherently assume IID data distribution. Our paper explicitly evaluates both IID and non-IID scenarios (see **Table 5 in Appendix C2.2**). Even under **highly heterogeneous** data distributions (specific experimental settings are detailed in our response to Reviewer vBZu's Q3), FedCEO consistently outperforms all baseline methods across different privacy configurations. Furthermore, as noted in our paper, the **parameter $\lambda$** can be adjusted to a relatively large value in heterogeneous settings to help each client **retain more personalized information** while still benefiting from collaborative learning.
> > >
> > > The core mechanism of semantic complementarity relies on intrinsic correlations within clients' semantic spaces, which persist even in non-IID scenarios. For example, clients specializing in different animal categories (e.g., cats vs. dogs) still share **low-level features** (edges, textures) that can be collaboratively enhanced through low-rank tensor optimization.
> > >
> > > > In Q3, you mentioned exploring alternative tensor optimization methods (e.g., deep tensor learning). Could these methods potentially leak client privacy due to additional parameters or nonlinear operations? How would you ensure compatibility with existing privacy constraints?
> > >
> > > Any alternative tensor optimization methods (e.g., deep tensor learning) must comply with the post-processing theorem of differential privacy (Theorem 4.5).
> > >
> > > For instance, if local DP employs Gaussian mechanisms, the tensor optimization method must satisfy:
> > > - **The network only processes already noised parameters (post-DP data)**.
> > > - **The tensor optimization model does not utilize Gaussian distribution priors during its formulation**.
> > >
> > > Additionally, we can leverage existing **gradient inversion attacks** (see Appendix C2.4) to empirically verify the privacy guarantees of the tensor-optimized model parameters, providing further validation of the framework's security.
> > >
> > > Thank you once again for your valuable feedback. Please feel free to let us know if you have any further questions.

---

### Official Review · Reviewer_RVpv · 2025-03-13

**Overall Recommendation:** 3

**Summary:**

The paper introduces FedCEO, a federated learning framework that aims to balance model utility and differential privacy by applying tensor low-rank proximal optimization (via T-tSVD) on noisy client parameters. The idea of smoothing the global semantic space is interesting, though its novelty is not entirely clear compared to existing spectral methods.

**Claims And Evidence:**

The authors claim an improved utility-privacy trade-off (O(√d)) and effective adaptation to different privacy settings.

**Essential References Not Discussed:**

There is room for a more comprehensive discussion on scalability in federated learning under differential privacy.

**Experimental Designs Or Analyses:**

The experimental section demonstrates that FedCEO can achieve competitive performance compared to existing methods. That said, the hyperparameter choices (such as λ and ϑ) are not thoroughly justified, leaving some questions about the robustness of the method in practical scenarios.

**Methods And Evaluation Criteria:**

The approach leverages tensor low-rank approximation to mitigate the impact of differential privacy noise. This method is sound.

**Other Comments Or Suggestions:**

Traditional tensor methods have been extensively studied, and it would be more convincing to try using them to validate the effectiveness of this method within deep learning approaches.

**Other Strengths And Weaknesses:**

Strengths:
1.The paper addresses a significant issue by proposing a way to mitigate DP noise effects using tensor methods.
2.The combination of theoretical analysis and experimental results is a strong point.

Weaknesses:
1.The paper lacks intuitive visualizations that could better demonstrate its noise robustness.
2. Experimental evaluation is limited to simple datasets (only two datasets), which may not fully reflect performance in real-world scenarios.
3. FedCEO seems to be a shallow method. What are its advantages and disadvantages compared to deep methods? It would be ideal to compare it with newer deep methods to highlight the value and significance of the research in this paper.

**Questions For Authors:**

See weaknesses.

**Relation To Broader Scientific Literature:**

The discussion does not fully differentiate FedCEO from related methods, leaving some ambiguity regarding its incremental contribution.

**Theoretical Claims:**

The paper includes comprehensive proofs for its main theoretical claims.

---

> ### Author Rebuttal · Authors · 2025-04-01
>
> We sincerely appreciate your constructive feedback and the opportunity to improve our paper. Below are our point-by-point responses to your comments:
>
> > **W1**
>
> **Figure 4 in the paper** visualizes the robustness of our method to different noise levels on CIFAR-10. Compared to other methods, our FedCEO (orange curve) exhibits less utility degradation as noise (privacy) intensity increases, demonstrating stronger utility-privacy trade-off and robustness to privacy noise (as reflected by the **minimal slope of the orange curve**, second only to non-private FedAvg (blue curve)). In the revised version, we plan to include visualizations on additional datasets to further strengthen this conclusion.
>
> > **W2**
>
> In addition to EMNIST and CIFAR-10 used in the main text, we have also evaluated FedCEO on **text data (Sent140)** with LSTM (see **Appendix C2.3 Table 6**), where our method consistently demonstrates superior performance.
>
> > **W3 & Other Comments/Suggestions**
>
> FedCEO is a novel federated learning parameter update framework that is **architecture-agnostic** (operating on parameter tensors) and compatible with deep architectures (e.g., AlexNet, see Table 6). Its key advantage lies in efficiency—it avoids the high computational costs of deep personalized methods like PPSGD and CENTAUR (see efficiency experiments in Appendix C2.1 Table 4).
>
> Furthermore, we have integrated the **deep tensor method CoSTCo [1]** (*modified for low-rank approximation task*) into our framework. Experimental results show that our approach (via T-tSVD) outperforms CoSTCo on CIFAR-10, likely because CoSTCo is more suited for sparse tensors and less compatible with noisy parameter tensors. Future work will explore deep tensor methods better aligned with FedCEO.
> | Dataset   | Model         | Setting ($\sigma_g$)  | FedCEO w/ CoSTCo  | FedCEO w/ T-tSVD |
> |----------|---------------|---------------|------------------|-----------------|
> |   |    |1.0  | 48.31% ± 0.9% | **54.16% ± 0.2%** |
> |    CIFAR-10       |     LeNet-5      | 1.5  | 44.70% ± 1.0% | **50.00% ± 0.5%** |
> |       |       | 2.0 |  32.25% ± 0.3% | **45.35% ± 0.9%** |
>
> [1] CoSTCo: A Neural Tensor Completion Model for Sparse Tensors (KDD 2019).
>
> > **Experimental Designs/Analyses**
>
> The initialization coefficient λ and geometric ratio ϑ are designed to **adaptively adjust the truncation threshold** $\frac{1}{2\lambda} \vartheta^{\frac{t}{I}}$ based on noise levels (Sec. 3.2). For stronger privacy guarantees (larger σg), smaller λ enhances smoothness (see Table 2), while ϑ > 1 accounts for accumulating DP noise during training. Detailed parameter selection guidelines are provided in Appendix C.1.
>
> Beyond the robustness analysis of parameter I in Table 3, we have added the following analysis for ϑ:
>
> | Dataset   | Model     | σ_g  | ϑ=1.00 | ϑ=1.01 | ϑ=1.02 | ϑ=1.03 | ϑ=1.04 | ϑ=1.05 | ϑ=1.06 | ϑ=1.07 | ϑ=1.08 | ϑ=1.09 | ϑ=1.10 |
> |-----------|-----------|------|--------|--------|--------|--------|--------|--------|--------|--------|--------|--------|--------|
> |   |                 | 1.0  | 50.09% | 53.62% | 53.85% | **54.16%** | 53.92% | 53.90% | 54.09% | 53.20% | 53.76% | 52.47% | 52.02% |
> | CIFAR-10  | LeNet-5    | 1.5  | 48.89% | 49.33% | 49.85% | 48.71% | 49.30% | 49.10% | 48.92% | **50.00%** | 49.62% | 48.94% | 48.28% |
> |           |           | 2.0  | 37.39% | 40.32% | 43.19% | 44.87% | **45.35%** | 44.50% | 44.81% | 41.32% | 43.90% | 39.75% | 40.27% |
>
> It can be observed that when ϑ > 1, the model performance shows significant improvement compared to ϑ = 1, while demonstrating **strong robustness to this parameter when ϑ > 1**. We will also include a **visual analysis of mixed λ-ϑ robustness** in the revised manuscript.
>
> > **Relation to Broader Literature**
>
> FedCEO introduces a **client-collaborative tensor low-rank optimization framework** for global model updates in DPFL, explicitly leveraging inter-client semantic complementarity in spectral space. Unlike prior spectral methods (e.g., CENTAUR [Shen et al., 2023] and [Jain et al., 2021]) that apply singular value decomposition (SVD) independently to client matrices, FedCEO stacks client parameters into a **higher-order tensor** and performs truncated tensor-SVD (T-tSVD). This enables adaptive truncation of high-frequency components across clients while preserving low-rank structures, thereby improving global model utility through inter-client information fusion.

---

> > ### Comment · Reviewer_RVpv · 2025-04-04
> >
> > After reviewing the authors' detailed responses, I am satisfied that my concerns have been addressed. The clarifications on the robustness of FedCEO (particularly the extensive evaluations on CIFAR-10, EMNIST, and Sent140) and the integration of additional experiments (such as the efficiency comparisons with methods like PPSGD and CENTAUR) convincingly demonstrate the method’s strengths. The added visualizations and discussions around the deep tensor method further solidify the paper’s contributions. So I am raising my score.

---

> > > ### Author Response · Authors · 2025-04-04
> > >
> > > Dear Reviewer,
> > >
> > > We are pleased to hear that your concerns have been addressed. Thank you for acknowledging our efforts! Your valuable suggestions are of great importance in improving the quality of our paper. If you have any further concerns, please feel free to let us know. We are more than happy to answer them for you.

---

### Official Review · Reviewer_KHNv · 2025-03-20

**Overall Recommendation:** 3

**Summary:**

The paper introduces a new method for FL training with DP guarantees. The authors argue that the method improves on existing work in terms of the utility-privacy trade-off, supporting their algorithm with a theoretical analysis and experiments. The method is based on a tensor low-rank proximal optimization of the (stacked) local parameters at the server and intuition is provided about why this truncates high-frequency components in spectral space.

**Claims And Evidence:**

The paper presents DP upper bounds on the epsilon, which are a usual in this field to evaluate the privacy guarantees of an algorithm. Experiments on EMNIST and CIFAR do present improvements in terms of accuracy, for a fixed noise level.

Some concerns are as follows:

- The theoretical improvement of the privacy-utility trade-off is stated for a joint objective which is a product of the utility improvement and the privacy guarantee. However, the paper does not discuss why this is a good way to merge the objectives, nor does it cite another work where such a discussion can be found. Why and when may the suggested product be preferable to alternative ways to mix the objectives, e.g. weighted average, or to a Pareto treatment, under which both objectives need to be improved for a method to be considered better?

- In the experiments, privacy is measured via the noise level $\sigma$ that is used. In general, the same $\sigma$ may lead to different levels of privacy for different learning algorithms. Why not use the guarantees provided by Theorems 4.4 and 4.5 (and similar) instead?

- Related to the last point, how do the guarantees in Lemma 4.4 and Theorem 4.5 compare?

**Essential References Not Discussed:**

NA

**Experimental Designs Or Analyses:**

The structure of the experiments make sense. However, on some of the experiments in Table 1, the gains compared to prior methods are relatively small. Estimates of standard deviation (under different random seeds) will be helpful here, in order to evaluate the gains from the proposed methods better.

**Methods And Evaluation Criteria:**

The way of evaluating the proposed methods: by theoretical upper bounds and experiments with varying noise, are to my awareness standard in this literature.

**Other Comments Or Suggestions:**

- Algorithm 2 pseudo code contains return commands. The current way I read this is that the algorithm will terminate after this, which is not the intended interpretation. I suggest the authors double-check that.

**Other Strengths And Weaknesses:**

NA

**Questions For Authors:**

See above.

**Relation To Broader Scientific Literature:**

The paper is well-positioned with respect to prior work, as it claims a clear contribution and discusses related work in detail.

**Theoretical Claims:**

I have not checked the proofs, but the assumptions and type of bounds are sensible.

---

> ### Author Rebuttal · Authors · 2025-04-01
>
> Thank you for your valuable feedback and for taking the time to provide thorough reviews! Below are our point-by-point responses to your comments:
>
> > **Claims And Evidence (Concern 1)**
>
> We appreciate the reviewer's suggestion.
> - The product form was chosen because it directly reflects the **joint tightness of the utility-privacy boundary**. This form highlights the intrinsic trade-off: improving one objective often requires relaxing the other.
> - Secondly, we maintain the **same** joint objective formulation ($\epsilon_u \cdot \epsilon_p$) as the prior SOTA method CENTAUR [1], enabling direct theoretical bound comparisons in our paper. Moreover, as shown in Theorems 4.3, 4.5, and Corollary 4.6, the product form eliminates the influence of the stochastic factor K.
> - Compared to weighted averaging [2] or Pareto optimization [3], the product form is more **concise and interpretable**. *We will add relevant discussions and citations in the revised version*.
>   - **Product vs. Weighted Average**: Weighted averaging (e.g., $\alpha \epsilon_u + (1-\alpha)\epsilon_{p}$) requires manual weight selection ($\alpha$), which may introduce subjectivity. The product form avoids weights and directly captures the coupling between objectives, making it more suitable for theoretical analysis. Additionally, weighted averaging is more susceptible to scale differences.
>   - **Product vs. Pareto Optimization**: Pareto requires simultaneous optimization of both objectives, but in practice, privacy and utility are often conflicting. The product form allows controlled compromise on one objective to achieve significant gains in the other, better aligning with practical needs.
>
> [1] "Share Your Representation Only: Guaranteed Improvement of the Privacy-Utility Tradeoff in Federated Learning" (ICLR2023)
>
> [2] "No free lunch theorem for security and utility in federated learning" (TIST2022)
>
> [3] "Optimizing privacy, utility, and efficiency in a constrained multi-objective federated learning framework" (TIST2024)
>
> > **Claims And Evidence (Concern 2)**
>
> For all compared methods in experiments, we adopted the **same DP mechanism** (user-level DP) and **unified implementation** (based on Opacus). Thus, their privacy guarantees are identical (i.e., same $\epsilon_{p}$). In the revision, we will supplement the quantitative relationship between $\sigma_g$ and $\epsilon_{p}$ (e.g., adding an $\epsilon_{p}$ column in Table 1) based on Theorem 4.5, demonstrating our method's superior utility under identical $(\epsilon_{p}, \delta)$.
>
> > **Claims And Evidence (Concern 3)**
>
> Lemma 4.4 provides the privacy guarantee for the baseline algorithm UDP-FedAvg, while Theorem 4.5 proves that FedCEO's low-rank optimization (a deterministic operation without noise-dependent priors) preserves privacy (post-processing immunity). Thus, their bounds align. We will explicitly emphasize this in the revision.
>
> > **Experimental Designs Or Analyses**
>
> Thank you for the suggestion. We will add standard deviations in the revision (e.g., FedCEO 78.05%±0.2 vs. CENTAUR 77.26%±0.3 for EMNIST at $\sigma_g=1.0$), confirming statistically significant improvements.
> | Dataset   | Model         | Setting ($\sigma_g$) | UDP-FedAvg       | PPSGD           | CENTAUR         | FedCEO  | FedCEO ($\vartheta>1$)     |
> |----------|---------------|---------------|------------------|-----------------|-----------------|-----------------|------------------|
> |  |  | 1.0| 76.59% ± 0.8% | 77.01% ± 0.4% | 77.26% ± 0.3% | 77.14% ± 0.5%   | **78.05% ± 0.2%** |
> | EMNIST | MLP-2-Layers | 1.5  | 69.91% ± 0.8% | 70.78% ± 0.6% | 71.86% ± 0.2%  | 71.56% ± 1.0%   | **72.44% ± 0.8%** |
> |  | | 2.0  | 60.32% ± 1.2% | 61.51% ± 1.1% | 62.12% ± 0.9% | 63.38% ± 0.7%   | **64.20% ± 0.6%** |
> |   |    |1.0 | 43.87% ± 2.1%  | 49.24% ± 0.9% | 50.14% ± 1.4% | 50.09% ± 0.5% | **54.16% ± 0.2%** |
> |    CIFAR-10       |     LeNet-5      | 1.5 | 34.34% ± 0.7%  | 47.56% ± 1.6% | 46.90% ± 0.9% | 48.89% ± 0.6% | **50.00% ± 0.5%** |
> |       |       | 2.0 | 26.88% ± 2.8%  | 34.61% ± 0.7% | 36.70% ± 2.4% | 37.39% ± 1.1% | **45.35% ± 0.9%** |
>
> > **Others**
>
> We will correct the usage of `return` in Algorithm 2 (e.g., replacing it with `yield` or annotations) and move key assumptions to the main text in future versions.

---

> > ### Comment · Reviewer_KHNv · 2025-04-02
> >
> > I thank the authors for their response, which clarifies most of my questions.
> >
> > However, I am still unsure why the product $\epsilon_u \epsilon_p$ is a better measure of performance compared to any other metric. The author say that the product avoids a hyperparameter, however one can equally consider $\epsilon_u^{\alpha} \epsilon_p^{\beta}$ and ask about how to select $\alpha$ and $\beta$.
> >
> > Has prior work argued why this is a good metric to look at?
> >
> > Can something be said about the improvements on the individual bounds on the utility and privacy? How do these individual bounds compare to prior work in the field?

---

> > > ### Author Response · Authors · 2025-04-06
> > >
> > > Thank you for your feedback. Regarding your questions about the trade-off objective formulation, we respond as follows:
> > >
> > > > However, I am still unsure why the product $\epsilon_u \epsilon_p$ is a better measure of performance compared to any other metric. The author say that the product avoids a hyperparameter, however one can equally consider $\epsilon_u^{\alpha} \epsilon_p^{\beta}$ and ask about how to select $\alpha$ and ask about how to select $\alpha$ and $\beta$.
> > >
> > > **Without introducing artificial weight priors** ($\alpha$ or $\beta$), the product form $\epsilon_u \cdot \epsilon_p$ as a trade-off objective demonstrates **greater robustness to the scale differences** of both utility and privacy metrics compared to the additive form $\epsilon_u + \epsilon_{p}$. $\epsilon_u \cdot \epsilon_p$ more **fairly** reflects their dynamic variations. The additive form primarily captures changes in the larger-valued metric while being insensitive to the other, thus requiring careful manual weight tuning. For example, when $\epsilon_{p} = 10$, if $\epsilon_u$ (utility loss) changes from 0.01 to 0.1 (a severe 10× utility degradation), the additive trade-off value would change by **less than 1%**. This makes it difficult to detect when the overall trade-off is disrupted.
> > >
> > > Furthermore, the product-form objective offers stronger interpretability that **aligns with practical optimization goals**. As illustrated in Figure 4, it corresponds geometrically to the area of rectangles formed by each $(\epsilon_p, \epsilon_u)$ data point. We will incorporate this analysis in the paper's subsequent version.
> > >
> > > We acknowledge that introducing weight parameters (to either additive or product forms) could help manually adjust the relative importance of different metrics (model utility versus user privacy). We sincerely appreciate this suggestion and will explore it in future work.
> > >
> > > > Has prior work argued why this is a good metric to look at?
> > >
> > > Prior work have thoroughly validated the effectiveness of the product metric, such as in Corollary 5.1 of CENTAUR [2]. For fair comparison, we followed their objective formulation in our paper and achieved at least a $\sqrt{d}$-order improvement over their utility-privacy trade-off bounds.
> > >
> > > > Can something be said about the improvements on the individual bounds on the utility and privacy? How do these individual bounds compare to prior work in the field?
> > >
> > > Certainly. For **utility**: Compared to [1] (Theorem 5.2), we (Theorem 4.3) achieve an **$O(d)$** improvement. Compared to [2] (Theorem 5.1), we achieve an **$O(\sqrt{d})$** improvement. This benefits from our method's flexible collaboration of semantic information across clients. For **privacy**: Our Theorem 4.5 guarantees $(\epsilon, \delta)$-user-level DP that is no weaker than previous methods.
> > >
> > > [1] "Differentially private model personalization" (NeurIPS 2021)
> > >
> > > [2] "Share Your Representation Only: Guaranteed Improvement of the Privacy-Utility Tradeoff in Federated Learning" (ICLR 2023)
> > >
> > > Thank you once again for your valuable insights, which will undoubtedly help us refine and strengthen our work. If you have any further concerns, we would be happy to discuss them with you!

---

### Decision · Program_Chairs · 2025-05-01

**Decision:**

Accept (poster)

**Comment:**

This paper introduces a federated learning framework (FedCEO) that aims to improve the utility-privacy trade-off in differentially private FL settings by promoting semantic "collaboration" among clients. The method applies tensor low-rank proximal optimization on stacked local model parameters at the server. This update helps filter out high-frequency noise components introduced by differential privacy mechanisms. Theoretical results show an improved utility-privacy trade-off bound scaling with sqrt(d). The empirical results on image and text datasets validate the framework’s effectiveness.

Several reviewers found the theoretical foundation solid. They also found the experiments well-structured. The framework's ability to outperform or match baseline methods on standard datasets such as CIFAR was also noted as a strong point by reviewers (despite these datasets have been used a lot). However, multiple reviewers raised concerns around the clarity and justification of certain methodological choices, especially regarding the definition and motivation of the utility-privacy tradeoff as a product objective (of epsilon values). There was also some concerns about insufficient discussion of hyperparameter selection, particularly the lack of experiments on more complex or large-scale datasets. In the rebuttal phase these concerns are (mostly) addressed and the reviewers generally tend toward acceptance of the paper.